# SAMPLING-BASED INFERENCE FOR LARGE LINEAR MODELS, WITH APPLICATION TO LINEARISED LAPLACE

**Javier Antorán**[*]
University of Cambridge

**Shreyas Padhy**[*]
University of Cambridge

**Riccardo Barbano**
University College London

**Eric Nalisnick**
University of Amsterdam

**David Janz**
University of Alberta

**José Miguel Hernández-Lobato**
University of Cambridge

## ABSTRACT

Large-scale linear models are ubiquitous throughout machine learning, with contemporary application as surrogate models for neural network uncertainty quantification; that is, the linearised Laplace method. Alas, the computational cost associated with Bayesian linear models constrains this method's application to small networks, small output spaces and small datasets. We address this limitation by introducing a scalable sample-based Bayesian inference method for conjugate Gaussian multi-output linear models, together with a matching method for hyperparameter (regularisation strength) selection. Furthermore, we use a classic feature normalisation method, the g-prior, to resolve a previously highlighted pathology of the linearised Laplace method. Together, these contributions allow us to perform linearised neural network inference with ResNet-18 on CIFAR100 (11M parameters, 100 output dimensions × 50k datapoints) and with a U-Net on a high-resolution tomographic reconstruction task (2M parameters, 251k output dimensions). An extended version of this work is available at `arxiv.org/pdf/2210.04994.pdf`.

## 1 INTRODUCTION

The linearised Laplace method, originally introduced by Mackay (1992), has received renewed interest in the context of uncertainty quantification for modern neural networks (NN) (Khan et al., 2019; Immer et al., 2021b; Daxberger et al., 2021a). The method constructs a surrogate Gaussian linear model for the NN predictions, and uses the error bars of that linear model as estimates of the NN's uncertainty. However, the resulting linear model is very large; the design matrix is sized number of parameters by number of datapoints times number of output classes. Thus, both the primal (weight space) and dual (observation space) formulations of the linear model are intractable. This restricts the method to small network or small data settings. Moreover, the method is sensitive to the choice of regularisation strength for the linear model (Immer et al., 2021a; Antorán et al., 2022c). Motivated by linearised Laplace, we study inference and hyperparameter selection in large linear models.

To scale inference and hyperparameter selection in Gaussian linear regression, we introduce a sample-based Expectation Maximisation (EM) algorithm. It interleaves E-steps, where we infer the model's posterior distribution over parameters given some choice of hyperparameters, and M-steps, where the hyperparameters are improved given the current posterior. Our contributions here are two-fold:

1. We enable posterior sampling for large-scale conjugate Gaussian-linear models with a novel sample-then-optimize objective, which we use to approximate the E-step.

2. We introduce a method for hyperparameter selection that requires only access to posterior samples, and not the full posterior distribution. This forms our M-step.

Combined, these allow us to perform inference and hyperparameter selection by solving a series of quadratic optimisation problems using iterative optimisation, and thus avoiding an explicit cubic cost in any of the problem's properties. Our method readily extends to non-conjugate settings, such as classification problems, through the use of the Laplace approximation. In the context of linearised

---

[*]Equal contribution. Correspondence to `ja666@cam.ac.uk` and `sp2058@cam.ac.uk` .

NNs, our approach also differs from previous work in that it avoids instantiating the full NN Jacobian matrix, an operation requiring as many backward passes as output dimensions in the network.

We demonstrate the strength of our inference technique in the context of the linearised Laplace procedure for image classification on CIFAR100 (100 classes × 50k datapoints) using an 11M parameter ResNet-18. We also consider a high-resolution (251k pixel) tomographic reconstruction (regression) task with a 2M parameter U-Net. In tackling these, we encounter a pathology in the M-step of the procedure first highlighted by Antorán et al. (2022c): the standard objective therein is ill-defined when the NN contains normalisation layers. Rather than using the solution proposed in Antorán et al. (2022c), which introduces more hyperparameters, we show that a standard feature-normalisation method, the g-prior (Zellner, 1986; Minka, 2000), resolves this pathology. For the tomographic reconstruction task, the regression problem requires a dual-form formulation of our E-step; interestingly, we show that this is equivalent to an optimisation viewpoint on Matheron's rule (Journel & Huijbregts, 1978; Wilson et al., 2020), a connection we believe to be novel.

## 2 Conjugate Gaussian regression and the EM algorithm

We study Bayesian conjugate Gaussian linear regression with multidimensional outputs, where we observe inputs $x_1, \ldots, x_n \in \mathbb{R}^d$ and corresponding outputs $y_1, \ldots, y_n \in \mathbb{R}^m$. We model these as

$$y_i = \phi(x_i)\theta + \eta_i, \tag{1}$$

where $\phi \colon \mathbb{R}^d \mapsto \mathbb{R}^m \times \mathbb{R}^{d'}$ is a known embedding function. The parameters $\theta$ are assumed sampled from $\mathcal{N}(0, A^{-1})$ with an unknown precision matrix $A \in \mathbb{R}^{d' \times d'}$, and for each $i \leq n, \eta_i \sim \mathcal{N}(0, B_i^{-1})$ are additive noise vectors with precision matrices $B_i \in \mathbb{R}^{m \times m}$ relating the $m$ output dimensions.

Our goal is to infer the posterior distribution for the parameters $\theta$ given our observations, under the setting of $A$ of the form $A = \alpha I$ for $\alpha > 0$ most likely to have generated the observed data. For this, we use the iterative procedure of Mackay (1992), which alternates computing the posterior for $\theta$, denoted $\Pi$, for a given choice of $A$, and updating $A$, until the pair $(A, \Pi)$ converge to a locally optimal setting. This corresponds to an EM algorithm (Bishop, 2006).

Henceforth, we will use the following stacked notation: we write $Y \in \mathbb{R}^{nm}$ for the concatenation of $y_1, \ldots, y_n$; $\mathrm{B} \in \mathbb{R}^{nm \times nm}$ for a block diagonal matrix with blocks $B_1, \ldots, B_n$ and $\Phi = [\phi(X_1)^T; \ldots; \phi(X_n)^T]^T \in \mathbb{R}^{nm \times d'}$ for the embedded design matrix. We write $M := \Phi^T \mathrm{B} \Phi$. Additionally, for a vector $v$ and a PSD matrix $G$ of compatible dimensions, $\|v\|_G^2 = v^T G v$.

With that, the aforementioned EM algorithm starts with some initial $A \in \mathbb{R}^{d' \times d'}$, and iterates:

- (E step) Given $A$, the posterior for $\theta$, denoted $\Pi$, is computed exactly as

$$\Pi = \mathcal{N}(\bar{\theta}, H^{-1}) \quad \text{where} \quad H = M + A \quad \text{and} \quad \bar{\theta} = H^{-1}\Phi^T \mathrm{B} Y. \tag{2}$$

- (M step) We lower bound the log-probability density of the observed data, i.e. the evidence, for the model with posterior $\Pi$ and precision $A'$ as (derivation in Appendix B.2)

$$\log p(Y; A') \geq -\tfrac{1}{2}\|\bar{\theta}\|_{A'}^2 - \tfrac{1}{2}\log\det(I + A'^{-1}M) + C =: \mathcal{M}(A'), \tag{3}$$

for C independent of $A'$. We choose an $A$ that improves this lower bound.

**Limited scalability**  The above inference and hyperparameter selection procedure for $\Pi$ and $A$ is futile when both $d'$ and $nm$ are large. The E-step requires the inversion of a $d' \times d'$ matrix and the M-step evaluating its log-determinant, both cubic operations in $d'$. These may be rewritten to instead yield a cubic dependence on $nm$ (as in Section 3.3), but under our assumptions, that too is not computationally tractable. Instead, we now pursue a stochastic approximation to this EM-procedure.

## 3 Evidence maximisation using stochastic approximation

We now present our main contribution, a stochastic approximation (Nielsen, 2000) to the iterative algorithm presented in the previous section. Our M-step requires only access to samples from $\Pi$. We introduce a method to approximate posterior samples through stochastic optimisation for the E-step.

## 3.1 Hyperparameter selection using posterior samples (M-step)

For now, assume that we have an efficient method of obtaining samples $\zeta_1, \ldots, \zeta_k \sim \Pi^0$ at each step, where $\Pi^0$ is a zero-mean version of the posterior $\Pi$, and access to $\bar{\theta}$, the mean of $\Pi$. Evaluating the first order optimality condition for $\mathcal{M}$ (see Appendix B.3) yields that the optimal choice of $A$ satisfies

$$\|\bar{\theta}\|_A^2 = \mathrm{Tr}\left\{H^{-1}M\right\} =: \gamma, \tag{4}$$

where the quantity $\gamma$ is the effective dimension of the regression problem. It can be interpreted as the number of directions in which the weights $\theta$ are strongly determined by the data. Setting $A = \alpha I$ for $\alpha = \gamma / \|\bar{\theta}\|^2$ yields a contraction step converging towards the optimum of $\mathcal{M}$ (Mackay, 1992).

Computing $\gamma$ directly requires the inversion of $H$, a cubic operation. We instead rewrite $\gamma$ as an expectation with respect to $\Pi^0$ using Hutchinson (1990)'s trick, and approximate it using samples as

$$\gamma = \mathrm{Tr}\left\{H^{-1}M\right\} = \mathrm{Tr}\left\{H^{-\frac{1}{2}}MH^{-\frac{1}{2}}\right\} = \mathbb{E}[\zeta_1^T M \zeta_1] \approx \tfrac{1}{k}\sum_{j=1}^k \zeta_j^T \Phi^T B\Phi \zeta_j =: \hat{\gamma}. \tag{5}$$

We then select $\alpha = \hat{\gamma}/\|\bar{\theta}\|^2$. We have thus avoided the explicit cubic cost of computing the log-determinant in the expression for $\mathcal{M}$ (given in (3)) or inverting $H$. Due to the block structure of B, $\hat{\gamma}$ may be computed in order $n$ vector-matrix products.

## 3.2 Sampling from the linear model's posterior using SGD (E-step)

Now we turn to sampling from $\Pi^0 = \mathcal{N}(0, H^{-1})$. It is known (also, shown in Appendix C.1) that for $\mathcal{E} \in \mathbb{R}^{nm}$ the concatenation of $\varepsilon_1, \ldots, \varepsilon_n$ with $\varepsilon_i \sim \mathcal{N}(0, B_i^{-1})$ and $\theta^0 \sim \mathcal{N}(0, A^{-1})$, the minimiser of the following loss is a random variable $\zeta$ with distribution $\Pi^0$:

$$L(z) = \tfrac{1}{2}\|\Phi z - \mathcal{E}\|_B^2 + \tfrac{1}{2}\|z - \theta^0\|_A^2. \tag{6}$$

This is called the "sample-then-optimise" method (Papandreou & Yuille, 2010). We may thus obtain a posterior sample by optimising this quadratic loss for a given sample pair $(\mathcal{E}, \theta^0)$. Examining $L$:

- The first term is data dependent. It corresponds to the scaled squared error in fitting $\mathcal{E}$ as a linear combination of $\Phi$. Its gradient requires stochastic approximation for large datasets.
- The second term, a regulariser centred at $\theta^0$, does not depend on the data. Its gradient can thus be computed exactly at every optimisation step.

Predicting $\mathcal{E}$, i.e. random noise, from features $\Phi$ is hard. Due to this, the variance of a mini-batch estimate of the gradient of $\|\Phi z - \mathcal{E}\|_B^2$ may be large. Instead, for $\mathcal{E}$ and $\theta^0$ defined as before, we propose the following alternative loss, equal to $L$ up to an additive constant independent of $z$:

$$L'(z) = \tfrac{1}{2}\|\Phi z\|_B^2 + \tfrac{1}{2}\|z - \theta^n\|_A^2 \quad \text{with} \quad \theta^n = \theta^0 + A^{-1}\Phi^T B\mathcal{E}. \tag{7}$$

The mini-batch gradients of $L'$ and $L$ are equal in expectation (see Appendix C.1). However, in $L'$, the randomness from the noise samples $\mathcal{E}$ and the prior sample $\theta^0$ both feature within the regularisation term—the gradient of which can be computed exactly—rather than in the data-dependent term. This may lead to a lower minibatch gradient variance. To see this, consider the variance of the single-datapoint stochastic gradient estimators for both objectives' data dependent terms. At $z \in \mathbb{R}^{d'}$, for datapoint index $j \sim \mathrm{Unif}(\{1, \ldots, n\})$, these are

$$\hat{g} = n\phi(X_j)^T(\phi(X_j)z - \varepsilon_j) \quad \text{and} \quad \hat{g}' = n\phi(X_j)^T\phi(X_j)z \tag{8}$$

for $L$ and $L'$, respectively. Direct calculation, presented in Appendix C.2, shows that

$$\tfrac{1}{n}\left[\mathrm{Var}\hat{g} - \mathrm{Var}\hat{g}'\right] = \mathrm{Var}(\Phi^T B\mathcal{E}) - 2\mathrm{Cov}(\Phi^T B\Phi z, \Phi^T B\mathcal{E}) =: \Delta. \tag{9}$$

Note that both $\mathrm{Var}\hat{g}$ and $\mathrm{Var}\hat{g}'$ are $d' \times d'$ matrices. We impose an order on these by considering their traces: we prefer the new gradient estimator $\hat{g}'$ if the sum of its per-dimension variances is lower than that of $\hat{g}$; that is if $\mathrm{Tr}\,\Delta > 0$. We analyse two key settings:

- At initialisation, taking $z = \theta^0$ (or any other initialisation independent of $\mathcal{E}$),

$$\mathrm{Tr}\,\Delta = \mathrm{Tr}\left\{\Phi^T B\mathbb{E}[\mathcal{E}\mathcal{E}^T]B\Phi\right\} - \mathrm{Tr}\left\{\Phi^T B\Phi\mathbb{E}[\theta^0\mathcal{E}^T]B\Phi\right\} = \mathrm{Tr}\left\{M\right\} > 0, \tag{10}$$

  where we used that $\mathbb{E}[\mathcal{E}\mathcal{E}^T] = B^{-1}$ and since $\mathcal{E}$ is zero mean and independent of $\theta^0$, we have $\mathbb{E}[\theta^0\mathcal{E}^T] = \mathbb{E}\theta^0\mathbb{E}\mathcal{E}^T = 0$. Thus, the new objective $L'$ is always preferred at initialisation.

- At convergence, that is, when $z = \zeta$, a more involved calculation presented in Appendix C.3 shows that $L'$ is preferred if

$$2\alpha\gamma > \operatorname{Tr} M. \tag{11}$$

  This is satisfied if $\alpha$ is large relative to the eigenvalues of $M$ (see Appendix C.4), that is, when the parameters are not strongly determined by the data relative to the prior.

When $L'$ is preferred both at initialisation and at convergence, we expect it to have lower variance for most minibatches throughout training. Even if the proposed objective $L'$ is not preferred at convergence, it may still be preferred for most of the optimisation, before the noise is fit well enough.

## 3.3 Dual form of E-step: Matheron's rule as optimisation

The minimiser of both $L$ and $L'$ is $\zeta = H^{-1}(A\theta^0 + \Phi^T B \mathcal{E})$. The dual (kernelised) form of this is

$$\zeta = \theta^0 + A^{-1}\Phi^T (B^{-1} + \Phi A^{-1}\Phi^T)^{-1}(\varepsilon - \Phi\theta^0), \tag{12}$$

which is known in the literature as Matheron's rule (Wilson et al., 2020, our Appendix D). Evaluating (12) requires solving a $mn$-dimensional quadratic optimisation problem, which may be preferable to the primal problem for $mn < d'$; however, this dual form cannot be minibatched over observations. For small $n$, we can solve this optimisation problem using iterative full-batch quadratic optimisation algorithms (e.g. conjugate gradients), significantly accelerating our sample-based EM iteration.

# 4 NN uncertainty quantification as linear model inference

Consider the problem of $m$-output prediction. Suppose that we have trained a neural network of the form $f \colon \mathbb{R}^{d'} \times \mathbb{R}^d \mapsto \mathbb{R}^m$, obtaining weights $\bar{w} \in \mathbb{R}^{d'}$, using a loss of the form

$$\mathcal{L}(f(w, \cdot)) = \sum_{i=1}^n \ell(y_i, f(w, x_i)) + \mathcal{R}(w) \tag{13}$$

where $\ell$ is a data fit term (a negative log-likelihood) and $\mathcal{R}$ is a regulariser. We now show how to use linearised Laplace to quantify uncertainty in the network predictions $f(\bar{w}, \cdot)$. We then present the g-prior, a feature normalisation that resolves a certain pathology in the linearised Laplace method when the network $f$ contains normalisation layers.

## 4.1 The linearised Laplace method

The linearised Laplace method consists of two consecutive approximations, the latter of which is necessary only if $\ell$ is non-quadratic (that is, if the likelihood is non-Gaussian):

1. We take a first-order Taylor expansion of $f$ around $\bar{w}$, yielding the surrogate model

$$h(\theta, x) = f(\bar{w}, x) + \phi(x)(\theta - \bar{w}) \ \text{ for } \ \phi(x) = \nabla_w f(\bar{w}, x). \tag{14}$$

   This is an affine model in the features $\phi(x)$ given by the network Jacobian at $x$.

2. We approximate the loss of the linear model $\mathcal{L}(h(\theta, \cdot))$ with a quadratic, and treat it as a negative log-density for the parameters $\theta$, yielding a Gaussian posterior of the form

$$\mathcal{N}(\bar{\theta}, \ (\nabla_\theta^2 \mathcal{L})^{-1}(h(\bar{\theta}, \cdot))) \quad \text{where} \quad \bar{\theta} \in \operatorname{argmin}_\theta \mathcal{L}(h(\theta, \cdot)). \tag{15}$$

   Direct calculation shows that $(\nabla_\theta^2 \mathcal{L})(h(\bar{\theta}, \cdot)) = (A + \Phi^T B \Phi) = H$, for $\nabla_w^2 \mathcal{R}(\bar{w}) = A$ and $B$ a block diagonal matrix with blocks $B_i = \nabla_{\hat{y}_i}^2 \ell(y_i, \hat{y}_i)$ evaluated at $\hat{y}_i = h(\bar{\theta}, x_i)$.

We have thus recovered a conjugate Gaussian multi-output linear model. We treat $A$ as a learnable parameter thereafter [1] In practice, we depart from the above procedure in two ways:

- We use the neural network output $f(\bar{w}, \cdot)$ as the predictive mean, rather than the surrogate model mean $h(\bar{\theta}, \cdot)$. Nonetheless, we still need to compute $\bar{\theta}$ to use within the M-step of the EM procedure. To do this, we minimise $\mathcal{L}(h(\theta, \cdot))$ over $\theta \in \mathbb{R}^{d'}$.

---

[1] The EM procedure from Section 2 is for the conjugate Gaussian-linear model, where it carries guarantees on non-decreasing model evidence, and thus convergence to a local optimum. These guarantees do not hold for non-conjugate likelihood functions, e.g., the softmax-categorical, where the Laplace approximation is necessary.

- We compute the loss curvature $B_i$ at predictions $\hat{y}_i = f(\bar{w}, x_i)$ in place of $h(\bar{\theta}, x_i)$, since the latter would change each time the regulariser $A$ is updated, requiring expensive re-evaluation.

Both of these departures are recommended within the literature (Antorán et al., 2022c).

### 4.2 ON THE COMPUTATIONAL ADVANTAGE OF SAMPLE-BASED PREDICTIONS

The linearised Laplace predictive posterior at an input $x$ is $\mathcal{N}(f(\bar{w}, x), \phi(x)H^{-1}\phi(x)^T)$. Even given $H^{-1}$, evaluating this naïvely requires instantiating $\phi(x)$, at a cost of $m$ vector-Jacobian products (i.e. backward passes). This is prohibitive for large $m$. However, expectations of any function $r : \mathbb{R}^m \mapsto \mathbb{R}$ under the predictive posterior can be approximated using only samples from $\Pi^0$ as

$$\mathbb{E}[r] \approx \tfrac{1}{k}\sum_{j=1}^{k} r(\psi_j) \quad \text{for} \quad \psi_j = f(\bar{w}, x) + \phi(x)\zeta_j \quad \text{with} \quad \zeta_1, \ldots, \zeta_k \sim \Pi^0, \tag{16}$$

requiring only $k$ Jacobian-vector products. In practice, we find $k$ much smaller than $m$ suffices.

### 4.3 FEATURE EMBEDDING NORMALISATION: THE DIAGONAL G-PRIOR

Due to symmetries and indeterminacies in neural networks, the embedding function $\phi(\cdot) = \nabla_w f(\bar{w}, \cdot)$ used in the linearised Laplace method yields features with arbitrary scales across the $d'$ weight dimensions. Consequently, the dimensions of the embeddings may have an (arbitrarily) unequal weight under an isotropic prior; that is, considering $\phi(x)\theta^0$ for $\theta^0 \sim \mathcal{N}(0, \alpha^{-1}I)$.

There are two natural solutions: either normalise the features by their (empirical) second moment, resulting in the normalised embedding function $\phi'$ given by

$$\phi'(x) = \phi(x)\operatorname{diag}(s) \quad \text{for} \quad s \in \mathbb{R}^{d'} \quad \text{given by} \quad s_i = [\Phi^T\mathrm{B}\Phi]_{ii}^{-1/2}, \tag{17}$$

or likewise scale the prior, setting $A = \alpha \operatorname{diag}((s_i^{-2})_{i=1}^{d'})$. The latter formulation is a diagonal version of what is known in the literature as the g-prior (Zellner, 1986) or scale-invariant prior (Minka, 2000).

The g-prior may, in general, improve the conditioning of the linear system. Furthermore, when the linearised network contains normalisation layers, such as batchnorm (that is, most modern networks), the g-prior is essential. Antorán et al. (2022c) show that normalisation layers lead to indeterminacies in NN Jacobians, that in turn lead to an ill-defined model evidence objective. They propose learning separate regularisation parameters for each normalised layer of the network. While fixing the pathology, this increases the complexity of model evidence optimisation. As we show in Appendix E, the g-prior cancels these indeterminacies, allowing for the use of a single regularisation parameter.

## 5 DEMONSTRATION: SAMPLE-BASED LINEARISED LAPLACE INFERENCE

We demonstrate our linear model inference and hyperparameter selection approach on the problem of estimating the uncertainty in NN predictions with the linearised Laplace method. First, in Section 5.1, we perform an ablation analysis on the different components of our algorithm using small LeNet-style CNNs trained on MNIST. In this setting, full-covariance Laplace inference (that is, exact linear model inference) is tractable, allowing us to evaluate the quality of our approximations. We then

---

**Algorithm 1:** Sampling-based linearised Laplace hyperparameter learning and inference

---

**Inputs:** initial $\alpha > 0$; $k, k' \in \mathbb{N}$, number of samples for stochastic EM and prediction, respectively.

Compute g-prior scaling vector $s$ as in (17)

Sample random regularisers $\theta_1^n, \ldots, \theta_k^n$ per (7)

**while** $\alpha$ *has not converged* **do**

    Find posterior mode $\bar{\theta}$ by optimising linear model loss $\mathcal{L}(h(\theta, \cdot))$, given in (13)

    Draw posterior samples $\zeta_1 \ldots \zeta_k$ by optimising objective $L'$ with $\theta_1^n, \ldots, \theta_k^n$

    Estimate effective dimension $\hat{\gamma}$, per (5), using samples $\zeta_1 \ldots \zeta_k$

    Update prior precision $\alpha \leftarrow \hat{\gamma}/\|\bar{\theta}\|_2^2$

Sample $k'$ random regularisers $\theta_1^{n'}, \ldots, \theta_{k'}^{n'}$ using optimised $\alpha$

Draw corresponding posterior samples $\zeta_1', \ldots, \zeta_{k'}'$ using loss $L'$

**Output:** posterior samples $\zeta_1', \ldots, \zeta_k'$

---

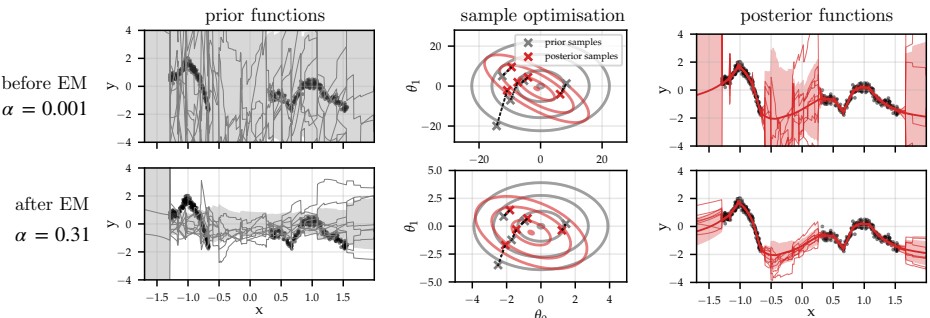

Figure 1: Illustration of our procedure for a fully connected NN on the toy dataset of Antorán et al. (2020). Top: prior function samples present large std-dev. (left). When these samples are optimised (middle shows a 2D slice of weight space), the resulting predictive errorbars are larger than the marginal target variance (right). Bottom: after EM, the std-dev. of prior functions roughly matches that of the targets (left), the overlap between prior and posterior is maximised, leading to shorter sample trajectories (center), and the predictive errorbars are qualitatively more appealing (right).

demonstrate our method at scale on CIFAR100 classification with a ResNet18 (Section 5.2) and the dual (kernelised) formulation of our method on tomographic image reconstruction using a U-Net (Section 5.3). We look at both marginal and joint uncertainty calibration and at computational cost.

For all experiments, our method avoids storing covariance matrices $H^{-1}$, computing their log-determinants, or instantiating Jacobian matrices $\phi(x)$, all of which have hindered previous linearised Laplace implementations. We interact with NN Jacobians only through Jacobian-vector and vector-Jacobian products, which have the same asymptotic computational and memory costs as a NN forward-pass (Novak et al., 2022). Unless otherwise specified, we use the diagonal g-prior and a scalar regularisation parameter. Algorithm 1 summarises our method, Figure 1 shows an illustrative example, and full algorithmic detail is in Appendix F. An implementation of our method in JAX can be found here. Additional experimental results are provided in Appendix H and Appendix I.

## 5.1 ABLATION STUDY: LENET ON MNIST

We first evaluate our approach on MNIST $m=10$ class image classification, where exact linearised Laplace inference is tractable. The training set consists of $n=60k$ observations and we employ 3 LeNet-style CNNs of increasing size: "LeNetSmall" ($d'=14634$), "LeNet" ($d'=29226$) and "LeNet-Big" ($d'=46024$). The latter is the largest model for which we can store the covariance matrix on an A100 GPU. We draw samples and estimate posterior modes using SGD with Nesterov momentum (full details in Appendix G). We use 5 seeds for each experiment, and report the mean and std. error.

**Comparing sampling objectives** We first compare the proposed objective $L'$ with the one standard in the literature $L$, using LeNet. The results are shown in Figure 2. We draw exact samples: $(\zeta_j^\star)_{j \leq k}$ through matrix inversion and assess sample fidelity in terms of normalised squared distance to these exact samples $\|\zeta - \zeta^\star\|_2^2 / \|\zeta^\star\|_2^2$. All runs share a prior precision of $\alpha \approx 5.5$ obtained with EM iteration. The effective dimension is $\hat{\gamma} \approx 1300$. Noting that the g-prior feature normalisation results in $\text{Tr}\, M = d'$, we can see that condition (11) is not satisfied ($2 \times 5.5 \times 1300 < 29k$). Despite this, the proposed objective converges to more accurate samples even when using a 16-times smaller batch size (left plot). The right side plots relate sample error to categorical symmetrised-KL (sym. KL) and logit Wasserstein-2 (W2) distance between the sampled and exact lin. Laplace predictive distributions on the test-set. Both objectives' prediction errors stop decreasing below a sample error of $\approx 0.5$ but, nevertheless, the proposed loss $L'$ reaches lower a prediction error.

**Fidelity of sampling inference** We compare our method's uncertainty using 64 samples against approximate methods based on the NN weight point-estimate (MAP), a diagonal covariance, and a KFAC estimate of the covariance (Martens & Grosse, 2015; Ritter et al., 2018) implemented with the Laplace library, in terms of similarity to the full-covariance lin. Laplace predictive posterior. As standard, we compute categorical predictive distributions with the probit approximation (Daxberger et al., 2021a). All methods use the same layerwise prior precision obtained with 5 steps of full-covariance EM iteration. For all three LeNet sizes, the sampled approximation presents the lowest categorical sym. KL and logit W2 distance to the exact lin. Laplace pred. posterior (Figure 3, LHS). The fidelity of competing approximations degrades with model size but that of sampling increases.

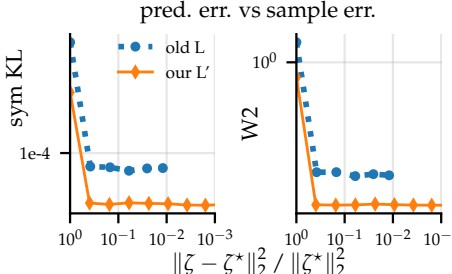

Figure 2: Left: optimisation traces for new and existing sampling losses averaged across 16 samples and 5 seeds. Right: for batch-size=1000, traces of sample-error vs distance to exact linear predictions.

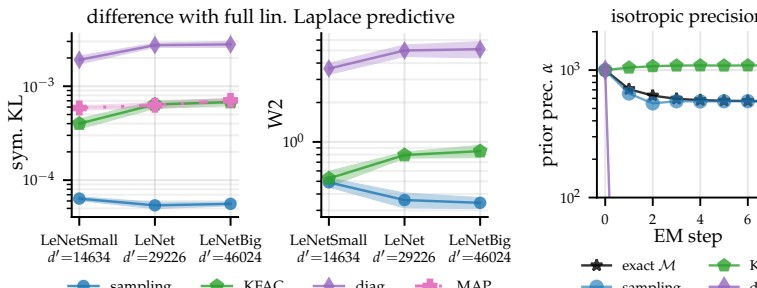

Figure 3: Left: similarity to exact lin. Laplace predictions on MNIST test-set for different approximate methods applied to NNs of increasing size. Centre right: comparison of EM convergence for a single hyperparameter across approximations. Right: layerwise convergence for exact and sampling methods.

**Accuracy of sampling hyperparameter selection** We compare our sampling EM iteration with 16 samples to full-covariance EM on LeNet without the g-prior. Figure 3, right, shows that for a single precision hyperparameter, both approaches converge in about 3 steps to the same value. In this setting, the diagonal covariance approximation diverges, and KFAC converges to a biased solution. We also consider learning layer-wise prior precisions by extending the M-step update from Section 3 to diagonal but non-isotropic prior precision matrices (see Appendix B.4). Here, neither the full covariance nor sampling methods converge within 15 EM steps. The precisions for all but the final layer grow, revealing a pathology of this prior parametrisation: only the final layer's Jacobian, i.e. the final layer activations, are needed to accurately predict the targets; other features are pruned away.

## 5.2 RESNET18 ON CIFAR100

We showcase the stability and performance of our approach by applying it to CIFAR100 $m=100$-way image classification. The training set consists of $n=50k$ observations, and we employ a ResNet-18 model with $d' \approx 11M$ parameters. We perform optimisation using SGD with Nesterov momentum and a linear learning rate decay schedule. Unless specified otherwise, we run 8 steps of EM with 6 samples to select $\alpha$. We then optimise 64 samples to be used for prediction. We run each experiment with 5 different seeds reporting mean and std. error. Full experimental details are in Appendix G.

**Stability and cost of sampling algorithm** Figure 4 shows that our sample-based EM converges in 6 steps, even when using a single sample. At convergence, $\alpha \approx 10^4$ and $\hat{\gamma} \approx 700$, so $2 \times 700 \times 10^4 =$

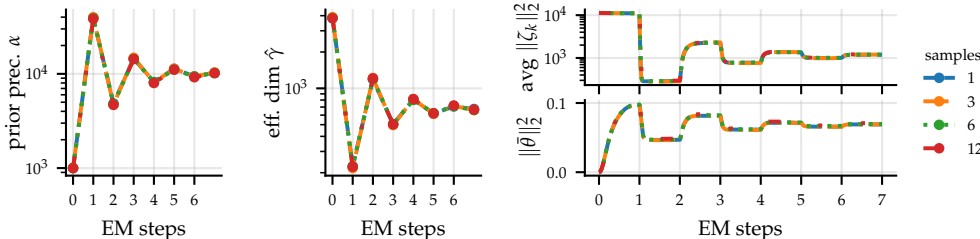

Figure 4: Left: prior precision optimisation traces for ResNet18 on CIFAR100 varying n. samples. Middle: same for the eff. dim. Right: average sample norm and posterior mean norm throughout successive EM steps' SGD runs while varying n. samples. Note that traces almost perfectly overlap.

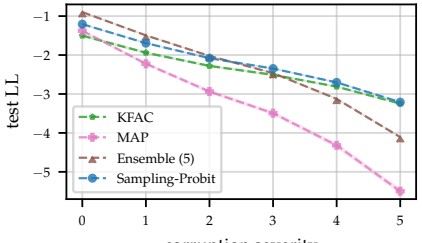

| | $\kappa$ | MAP | Ensemble (5) | KFAC | Sampling |
|---|---|---|---|---|---|
| marginal LL | 1 | $-1.40 \pm 0.00$ | $\mathbf{-0.90 \pm 0.00}$ | $-1.12 \pm 0.01$ | $-1.07 \pm 0.01$ |
| joint LL | 2 | $-13.97 \pm 0.01$ | $-6.86 \pm 0.01$ | $\mathbf{-4.92 \pm 0.04}$ | $-5.14 \pm 0.04$ |
| | 3 | $-27.89 \pm 0.03$ | $-14.17 \pm 0.03$ | $-10.83 \pm 0.12$ | $\mathbf{-10.77 \pm 0.09}$ |
| | 4 | $-41.83 \pm 0.03$ | $-22.29 \pm 0.04$ | $-19.02 \pm 0.22$ | $\mathbf{-18.04 \pm 0.18}$ |
| | 5 | $-55.89 \pm 0.02$ | $-31.07 \pm 0.09$ | $-29.40 \pm 0.40$ | $\mathbf{-26.75 \pm 0.26}$ |

Figure 5: Performance under distribution shift for ResNet18 and CIFAR100.

Table 1: Comparison of methods' marginal and joint prediction performance for ResNet18 on CIFAR100.

$1.4 \times 10^7 > 1.1 \times 10^7$. Thus, (11) is satisfied and $L'$ is preferred. We use 50 epochs of optimisation for the posterior mode and 20 for sampling. When using 2 samples, the cost of one EM step with our method is 45 minutes on an A100 GPU; for the KFAC approximation, this takes 20 minutes.

**Evaluating performance in the face of distribution shift**    We employ the standard benchmark for evaluating methods' test Log-Likelihood (LL) on the increasingly corrupted data sets of Hendrycks & Gimpel (2017); Ovadia et al. (2019). We compare the predictions made with our approach to those from 5-element deep ensembles, arguably the strongest baseline for uncertainty quantification in deep learning (Lakshminarayanan et al., 2017; Ashukha et al., 2020), with point-estimated predictions (MAP), and with a KFAC approximation of the lin. Laplace covariance (Ritter et al., 2018). For the latter, constructing full Jacobian matrices for every test point is computationally intractable, so we use 64 samples for prediction, as suggested in Section 4.2. The KFAC covariance structure leads to fast log-determinant computation, allowing us to learn layer-wise prior precisions (following Immer et al., 2021a) for this baseline using 10 steps of non-sampled EM. For both lin. Laplace methods, we use the standard probit approximation to the categorical predictive (Daxberger et al., 2021b). Figure 5 shows that for in-distribution inputs, ensembles performs best and KFAC overestimates uncertainty, degrading LL even relative to point-estimated MAP predictions. Conversely, our method improves LL. For sufficiently corrupted data, our approach outperforms ensembles, also edging out KFAC, which fares well here due to its consistent overestimation of uncertainty.

**Joint predictions**    Joint predictions are essential for sequential decision making, but are often ignored in the context of NN uncertainty quantification (Janz et al., 2019). To address this, we replicate the "*dyadic sampling*" experiment proposed by Osband et al. (2022). We group our test-set into sets of $\kappa$ data points and then uniformly re-sample the points in each set until sets contain $\tau$ points. We then evaluate the LL of each set jointly. Since each set only contains $\kappa$ distinct points, a predictor that models self-covariances perfectly should obtain an LL value at least as large as its marginal LL for all values of $\kappa$. We use $\tau = 10(\kappa - 1)$ and repeat the experiment for 10 test-set shuffles. Our setup remains the same as above but we use Monte Carlo marginalisation instead of probit, since the latter discards covariance information. Table 1 shows that ensembles make calibrated predictions marginally but their joint predictions are poor, an observation also made by Osband et al. (2021). Our approach is competitive for all $\kappa$, performing best in the challenging large $\kappa$ cases.

## 5.3    TOMOGRAPHIC RECONSTRUCTION

To demonstrate our approach in dual (kernelised) form, we replicate the setting of Barbano et al. (2022a;b) and Antorán et al. (2022b), where linearised Laplace is used to estimate uncertainty for a tomographic reconstruction outputted by a U-Net autoencoder. We provide an overview of the problem in Appendix G.3, but refer to Antorán et al. (2022b) for full detail. Whereas the authors use a single EM step to learn hyperparameters, we use our sample-based variant and run 5 steps. Unless otherwise specified, we use 16 samples for stochastic EM, and 1024 for prediction.

We test on the real-measured $\mu$CT dataset of $251k$ pixel scans of a single walnut released by Der Sarkissian et al. (2019b). We train $d' = 2.97M$ parameter U-Nets on the $m = 7680$ dimensional observation used by Antorán et al. (2022b) and a twice as large setting $m = 15360$. Here, the U-Net's input is clamped to a constant ($n = 1$), and its parameters are optimised to output the reconstructed image. As a result, we do not need mini-batching and can draw samples using Matheron's rule (12). We solve the linear system contained therein using conjugate gradient (CG) iteration implemented

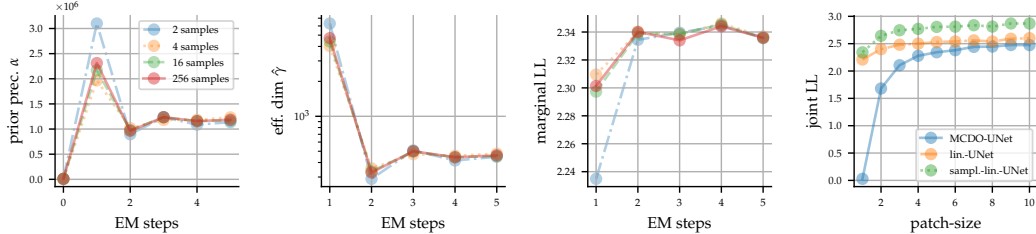

Figure 6: Left 3 plots: traces of prior precision, eff. dim., and marginal test LL vs EM steps for the tomographic reconstruction task with $m = 7680$. Right: joint test LL for varying image patch sizes.

Table 2: Tomographic reconstruction: test LL and wall-clock times (A100 GPU) for both data sizes.

| | $m = 7680$ | | | | $m = 15360$ | | | |
|---|---|---|---|---|---|---|---|---|
| | LL | | wall-clock time (min.) | | LL | | wall-clock time (min.) | |
| Method | marginal | $(10 \times 10)$ | params optim. | prediction | marginal | $(10 \times 10)$ | params. optim. | prediction |
| MCDO-UNet | 0.028 | 2.474 | 0 | $3'$ | 0.002 | 2.762 | 0 | $3'$ |
| lin.-UNet | 2.214 | 2.601 | $1260'$ | $196'$ | – | – | – | – |
| sampl.-lin.-UNet | **2.341** | **2.869** | $12'$ | $14'$ | **2.310** | **2.972** | $15'$ | $14'$ |

with GPyTorch (Gardner et al., 2018). We accelerate CG with a randomised SVD preconditioner of rank 400 (alg. 5.6 in Halko et al., 2011). See Appendix G for full experimental details.

**Stability and cost of sampling algorithm**    Figure 6 shows that sample-based EM iteration converges within 4 steps using as few as 2 samples. Table 2 shows the time taken to perform 5 sample-based EM steps for both the $m{=}7680$ and $m{=}15360$ settings; avoiding explicit estimation of the covariance log-determinant provides us with a two order of magnitude speedup relative to Antorán et al. (2022c) for hyperparameter learning. By avoiding covariance inversion, we obtain an order of magnitude speedup for prediction. Furthermore, while scaling to double the observations $m{=}15360$ is intractable with the previous method, our sampling method requires only a $25\%$ increase in computation time.

**Predictive performance**    Figure 7 shows, qualitatively, that the marginal standard deviation assigned to each pixel by our method aligns with the pixelwise error in the U-Net reconstruction in a fine-grained manner. By contrast, MC dropout (MCDO), the most common baseline for NN uncertainty estimation in tomographic reconstruction (Laves et al., 2020; Tölle et al., 2021), spreads uncertainty more uniformly across large sections of the image. Table 2 shows that the pixelwise LL obtained with our method exceeds that obtained by Antorán et al. (2022b), potentially due to us optimising the prior precision to convergence while the previous work could only afford a single EM step. The rightmost plot in Figure 6 displays joint test LL, evaluated on patches of neighbouring pixels. Our method performs best. MCDO's predictions are poor marginally. They improve when considering covariances, although remaining worse than lin. Laplace.

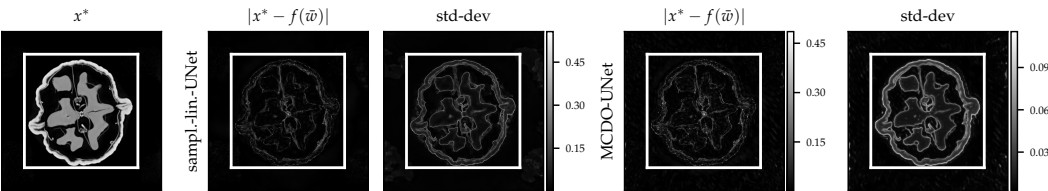

Figure 7: Original $501{\times}501$ pixel walnut image and reconstruction error for a $m{=}7680$ dimensional observation, along with pixel-wise std-dev obtained with sampling lin. Laplace and MCDO.

## 6    CONCLUSION

Our work introduced a sample-based approximation to inference and hyperparameter selection in Gaussian linear multi-output models. This allowed us to scale the linearised Laplace method to ResNet-18 on CIFAR100, where it was computationally intractable with existing methods. We also demonstrated the strength of the approach on a high resolution tomographic reconstruction task, where it decreases the cost of hyperparameter selection by two orders of magnitude. The uncertainty estimates obtained through our method are well-calibrated not just marginally, but also jointly across predictions. Thus, our work may be of interest in the fields of active and reinforcement learning, where joint predictions are of importance, and computation of posterior samples is often needed.

## REPRODUCIBILITY STATEMENT

In order to aid the reproduction of our results, we provide a high-level overview of our procedure in algorithm 1 and the fully detailed algorithms we use in our two major experiments in Appendix F. Appendix G provides full experimental details for all datasets and models used in our experiments. Our code is available in a repository at `this link`.

## ACKNOWLEDGEMENTS

The authors would like to thank Alex Terenin and Marine Schimel for helpful discussions. JA acknowledges support from Microsoft Research, through its PhD Scholarship Programme, and from the EPSRC. SP acknowledges support from the Harding Distinguished Postgraduate Scholars Programme Leverage Scheme. JMHL acknowledges support from a Turing AI Fellowship under grant EP/V023756/1. This work has been performed using resources provided by the Cambridge Tier-2 system operated by the University of Cambridge Research Computing Service (http://www.hpc.cam.ac.uk) funded by an EPSRC Tier-2 capital grant. This work was also supported with Cloud TPUs from Google's TPU Research Cloud (TRC).

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

## A    RELATED WORK

**Bayesian Gaussian linear models**   This work builds on the rich literature of Bayesian linear regression (Gull, 1989; Bishop, 2006; Rasmussen & Williams, 2006). Specifically, we present a stochastic approximation to the iterative algorithm for hyperparameter selection introduced by (Mackay, 1992) and extended by Tipping (2001); Tipping & Faul (2003); Wipf & Nagarajan (2007); Antorán et al. (2022c). Analytical tractability makes linear models ubiquitous in machine learning, with applications in genomics (Runcie et al., 2021), reinforcement learning (Ash et al., 2022), and pandemic modelling (Nicholson et al., 2022), among others. Alas, linear models are held back by a cost of inference cubic in the number of parameters when expressed in primal form, or cubic in the number of observations for the dual (i.e. kernelised or Gaussian Process) form. Additionally, for non-Gaussian likelihoods, e.g. in classification, inference is no longer closed form. The most common approximations used in these settings are Laplace's method (Mackay, 1992) and variational inference (Hensman et al., 2013). Khan et al. (2019) and Adam et al. (2021) show that every Gaussian approximation corresponds to the true posterior of a surrogate regression problem with the same features, a fact which we use in this work to apply sample-then-optimise to Laplace posteriors.

**Sample-then-optimise**   Papandreou & Yuille (2010); de G. Matthews et al. (2017) phrase sampling from a conjugate Gaussian-linear model as solving a perturbed quadratic optimisation problem. This method has been applied for uncertainty estimation in non-linearised NNs by Osband et al. (2018; 2021), and Pearce et al. (2020), although in these setting it does not draw exact posterior samples. In this work, we show sample-then-optimise to be the primal form of Matheron's rule (Journel & Huijbregts, 1978; Hoffman & Ribak, 1991), a method for updating jointly Gaussian samples into conditional samples, which was recently repopularised by Wilson et al. (2020).

**Linearised neural networks**   Introduced by Mackay (1992), this approximation yields closed-form errorbars for Laplace posteriors. Lawrence (2000) and Ritter et al. (2018) found the Laplace approximation to underperform without the linearisation step. Khan et al. (2019) and Immer et al. (2021b) re-popularised the linearisation step by showing that it improves the quality of uncertainty estimates. Kristiadi et al. (2020) show that the Laplace approximation is sufficient to resolve certain pathologies of point-estimated NNs' predictions. Immer et al. (2021a) and Antorán et al. (2022a;c) explore the linear model's evidence for model selection. Immer et al. (2022) shows the objective can even be used to learn invariances in deep models. Daxberger et al. (2021b) and Maddox et al. (2021) introduce subnetwork and finite differences approaches, respectively, for faster inference with the linearised model. This line of work is also closely related to the neural tangent kernel (Jacot et al., 2018; Lee et al., 2019; Novak et al., 2020) in which NNs are linearised at initialisation.

**The g-prior**, originally introduced by Zellner (1986), consists of a centred Gaussian with covariance matching the inverse of the Fisher information matrix. Resultantly, the g-prior ensures inferences are independent of the units of measurement of the covariates (Minka, 2000). Since then, it has extensively used in the context of model selection for generalised linear models (Liang et al., 2008; Bové & Held, 2011; Baragatti & Pommeret, 2012). In the large-scale setting, we overcome the computational intractability of the Fisher by diagonalising the g-prior while preserving its scale-invariance property.

## B    MODEL EVIDENCE LOWER BOUND AND THE EFFECTIVE DIMENSION

### B.1    EQUIVALENT FORMULATIONS OF EFFECTIVE DIMENSION

We begin by relating two standard forms of effective dimension, which we use throughout. Starting with the form standard in the kernel-based literature (that without an explicit $d'$ dependence),

$$\gamma = \text{Tr}\left\{(A+M)^{-1}M\right\} = \text{Tr}\left\{(I+A^{-1}M)^{-1}A^{-1}M\right\} = \text{Tr}\left\{I - (I+A^{-1}M)^{-1}\right\} \quad (18)$$

$$= d' - \text{Tr}\left\{A(A+M)^{-1}\right\}, \quad (19)$$

we have arrived at the form used within the finite-dimensional linear modelling literature (Mackay, 1992; Wipf & Nagarajan, 2007; Maddox et al., 2020).

## B.2 Derivation of $\mathcal{M}$ as a lower bound on the model evidence

Let $p_\theta$ be the Lebesgue density of $\mathcal{N}(\Phi\theta, B^{-1})$, $P = \mathcal{N}(0, A'^{-1})$ and $Q = \mathcal{N}(\bar{\theta}, (M + A')^{-1})$. Then,

$$\log p(Y; A') = \log \int p_\theta(Y) dP = \log \int p_\theta(Y) \frac{dP}{dQ} dQ \geq \int \log \left[ p_\theta(Y) \frac{dP}{dQ} \right] dQ \tag{20}$$

$$= \int \log p_\theta(Y) dQ - \mathrm{D}(Q\|P). \tag{21}$$

where D denotes the KL-divergence. Starting with the first term,

$$\int \log p_\theta(Y) dQ = \frac{1}{2} \int -n \log 2\pi + \log \det \mathrm{B} - (Y - \Phi\theta)^T \mathrm{B}(Y - \Phi\theta) dQ \tag{22}$$

$$= \frac{1}{2} \left[ -n \log 2\pi + \log \det \mathrm{B} \right] - \frac{1}{2} \int (Y - \Phi\theta)^T \mathrm{B}(Y - \Phi\theta) dQ, \tag{23}$$

and expanding the quadratic form,

$$\int (Y - \Phi\theta)^T \mathrm{B}(Y - \Phi\theta) dQ = Y^T \mathrm{B} Y - 2Y^T \mathrm{B}\Phi \int \theta dQ + \int \theta^T \Phi^T \mathrm{B}\Phi\theta dQ \tag{24}$$

$$= Y^T \mathrm{B} Y - 2Y^T \mathrm{B}\Phi\bar{\theta} + \int \theta^T M\theta dQ. \tag{25}$$

To handle the final integral, consider that

$$\gamma = \mathrm{Tr}\left\{ M(M + A')^{-1} \right\} \tag{26}$$

$$= \mathrm{Tr}\left\{ M \int (\theta - \bar{\theta})(\theta - \bar{\theta})^T dQ \right\} \tag{27}$$

$$= -\mathrm{Tr}\left\{ M\bar{\theta}\bar{\theta}^T \right\} + \mathrm{Tr}\left\{ M \int \theta\theta^T dQ \right\} \tag{28}$$

$$= -\bar{\theta}^T M\bar{\theta} + \int \theta^T M\theta dQ, \tag{29}$$

and thus

$$\int \log p_\theta(Y) dQ = \frac{1}{2} \left[ \log \det \mathrm{B} - n \log 2\pi - (Y - \Phi\bar{\theta})^T \mathrm{B}(Y - \Phi\bar{\theta}) - \gamma \right] \tag{30}$$

$$= \log p_{\bar{\theta}}(Y) - \frac{1}{2}\gamma. \tag{31}$$

The KL between two multivariate Gaussians is a standard result, yielding

$$\mathrm{D}(Q\|P) = \frac{1}{2} \left[ -\log \det A' + \log \det(M + A') - d' + \bar{\theta}^T A'\bar{\theta} + \mathrm{Tr}\left\{ A'(M + A')^{-1} \right\} \right] \tag{32}$$

$$= \frac{1}{2} \left[ -\log \det A' + \log \det(M + A') + \bar{\theta}^T A'\bar{\theta} - \gamma \right], \tag{33}$$

where we used that $\gamma = d' - \mathrm{Tr}\left\{ A'(M + A')^{-1} \right\}$.

Putting together (31) and (33), we obtain

$$\log p(Y; A') \geq \log p_{\bar{\theta}}(Y) - \frac{1}{2} \log \det(A'^{-1}M + I) - \frac{1}{2}\|\bar{\theta}\|_{A'}^2 = \mathcal{M}(A'), \tag{34}$$

which is the stated result up to taking $C = \log p_{\bar{\theta}}(Y)$.

## B.3 First order optimality condition for $\mathcal{M}$

Consider the derivative of $\mathcal{M}$. We have,

$$\nabla\mathcal{M}(A) = -\frac{1}{2} \left[ \nabla\|\bar{\theta}\|_A^2 + \nabla \log \det(A + M) - \nabla \log \det A \right], \tag{35}$$

where we expanded $\log \det(I + A^{-1}M) = \log \det(A + M) - \log \det A$. Taking the respective derivatives and setting equal to zero at $A$, this leads to the condition

$$\bar{\theta}\bar{\theta}^T = (I - (I + A^{-1}M)^{-1})A^{-1}. \tag{36}$$

Post-multiplying by $A$ and applying the push-through identity, we obtain

$$\bar{\theta}\bar{\theta}^T A = M(A + M)^{-1}. \tag{37}$$

For the above to hold, it is necessary that the traces of both sides are equal. Thus,

$$\|\bar{\theta}\|_A^2 = \mathrm{Tr}\{\bar{\theta}\bar{\theta}^T A\} = \mathrm{Tr}\{M(A + M)^{-1}\} = \gamma, \tag{38}$$

which is the stated first order optimality condition, up to a cyclic permutation.

### B.4 M-step for feature-wise regularisation strengths

We can leverage the primal form expression for the effective dimension given in Appendix B.1 to extend the above first order optimality condition to the feature-wise regulariser setting.

Consider a sub-vector of our weight vector contiguous between the $i$th and $j$th weights written as $\bar{\theta}_{i:j}$. Note that we only choose contiguous weights for notational convenience but it is not necessary to do so in general.

The first order condition from Appendix B.3 is satisfied if for any $i, j$ with $i < j$ we have

$$\alpha\|\bar{\theta}_{i:j}\|^2 = j - i - \sum_{k=i}^{j}[A]_{kk}[(A + M)^{-1}]_{kk} \coloneqq \gamma_{i:j}. \tag{39}$$

We assume $[A]_{kk} = \alpha$ for all $i \le k < j$. Thus, we may update the regulariser for each separate weight sub-vector as $\alpha = \gamma_{i:j}/\|\bar{\theta}_{i:j}\|^2$.

## C   Analysis of losses and loss gradient estimator variances

### C.1   On loss minima

The losses $L$ and $L'$ are strictly convex, thus to confirm they have the same unique minimum, it suffices to consider the respective first order optimality conditions, $\nabla L(\zeta) = 0$ and $\nabla L'(\zeta') = 0$. We have,

$$\nabla L(\zeta) = \Phi^T \mathrm{B}(\Phi\zeta - \mathcal{E}) + A(\zeta - \theta^0), \tag{40}$$

and

$$\nabla L'(\zeta') = \Phi^T \mathrm{B}\Phi\zeta' + A(\zeta' - A^{-1}\Phi^T \mathrm{B}\mathcal{E} - \theta^0) \tag{41}$$

$$= \Phi^T \mathrm{B}(\Phi\zeta' - \mathcal{E}) + A(\zeta' - \theta^0) \tag{42}$$

Thus $\zeta = \zeta'$ almost surely. Moreover, $L'(z) = L(z) + C$ for all $z$, for $C$ a constant independent of $z$.

To determine the distribution of $\zeta$, note that it is a linear transformation of zero-mean Gaussian random variables, and thus itself a zero-mean Gaussian random variable. Rearranging the first order optimality condition, we find that

$$\zeta = H^{-1}(\Phi^T \mathrm{B}\mathcal{E} + A\theta^0). \tag{43}$$

Thus

$$\mathbb{E}[\zeta\zeta^T] = H^{-1}\mathbb{E}[(\Phi^T \mathrm{B}\mathcal{E} + A\theta^0)(\Phi^T \mathrm{B}\mathcal{E} + A\theta^0)^T]H^{-1} \tag{44}$$

$$= H^{-1}\left(\Phi^T \mathrm{B}\mathbb{E}[\mathcal{E}\mathcal{E}^T]\mathrm{B}\Phi + A\mathbb{E}[\theta^0\theta^0]A + 2\Phi^T \mathrm{B}\mathbb{E}[\mathcal{E}(\theta^0)^T]A\right)H^{-1} \tag{45}$$

$$= H^{-1}(\Phi^T \mathrm{B}\Phi + A)H^{-1} = H^{-1}HH^{-1} = H^{-1}. \tag{46}$$

And so $\zeta \sim \mathcal{N}(0, H^{-1}) = \Pi^0$ as claimed.

## C.2 Loss gradient variance condition

Taking $j \sim \text{Unif}(\{1, \ldots, n\})$, the gradient estimators for the data-dependent terms of $L$ and $L'$ are

$$\hat{g} = n\nabla\|\phi(x_j)z - \varepsilon_j\|_{B_j}^2 = n\phi(x_j)^T B_j(\phi(x_j)z - \varepsilon_j) \tag{47}$$

and

$$\hat{g}' = n\nabla\|\phi(x_j)z\|_{B_j}^2 = n\phi(x_j)^T B_j\phi(x_j)z, \tag{48}$$

respectively. Their variances are related as

$$\text{Var}(\hat{g}) = \text{Var}(n\phi(x_j)^T B_j(\phi(x_j)z - \varepsilon_j)) \tag{49}$$

$$= \text{Var}(n\phi(x_j)^T B_j\phi(x_j)z) + \text{Var}(n\phi(x_j)^T B_j\varepsilon_j)$$

$$- 2\text{Cov}(n\phi(x_j)^T B_j\phi(x_j)z, \ n\phi(x_j)^T B_j\varepsilon_j) \tag{50}$$

$$= \text{Var}(\hat{g}') + \text{Var}(n\phi(x_j)^T B_j\varepsilon_j) - 2\text{Cov}(n\phi(x_j)^T B_j\phi(x_j)z, \ n\phi(x_j)^T B_j\varepsilon_j) \tag{51}$$

Evaluating the variance and covariance, we have

$$\text{Var}\left(n\phi(x_j)^T B_j\varepsilon_j\right) = n\text{Var}(\Phi^T \text{B}\mathcal{E}) \tag{52}$$

and

$$\text{Cov}(n\phi(x_j)^T B_j\phi(x_j)z, \ n\phi(x_j)^T B_j\varepsilon_j) = n\text{Cov}(\Phi^T \text{B}\Phi z, \ \Phi^T \text{B}\mathcal{E}), \tag{53}$$

and thus

$$\text{Var}\hat{g} - \text{Var}\hat{g}' = n\left[\text{Var}(\Phi^T \text{B}\mathcal{E}) - 2\text{Cov}(\Phi^T \text{B}\Phi z, \Phi^T \text{B}\mathcal{E})\right] =: n\Delta. \tag{54}$$

## C.3 Condition at convergence

Now consider $\text{Tr}\,\Delta$ for $z = \zeta \sim \Pi^0$, the optimum of both $L$ and $L'$. From the first order optimality condition,

$$\zeta = H^{-1}(\Phi^T \text{B}\mathcal{E} + A\theta^0). \tag{55}$$

Proceeding to rearrange the condition at $z = \zeta$,

$$\text{Tr}\,\Delta = \text{Tr}\left\{\mathbb{E}\Phi^T \text{B}\mathcal{E}(\Phi^T \text{B}\mathcal{E} - 2\Phi^T \text{B}\Phi\zeta)^T\right\} \tag{56}$$

$$= \text{Tr}\left\{\mathbb{E}\Phi^T \text{B}\mathcal{E}(\Phi^T \text{B}\mathcal{E} - 2\Phi^T \text{B}\Phi H^{-1}(\Phi^T \text{B}\mathcal{E} + A\theta^0))^T\right\} \tag{57}$$

$$= \text{Tr}\left\{\mathbb{E}\Phi^T \text{B}\mathcal{E}(\Phi^T \text{B}\mathcal{E} - 2\Phi^T \text{B}\Phi H^{-1}(\Phi^T \text{B}\mathcal{E} + A\mathbb{E}[\theta^0]))^T\right\} \tag{58}$$

$$= \text{Tr}\left\{\mathbb{E}\Phi^T \text{B}\mathcal{E}(\Phi^T \text{B}\mathcal{E} - 2\Phi^T \text{B}\Phi H^{-1}\Phi^T \text{B}\mathcal{E})^T\right\}, \tag{59}$$

$$= \text{Tr}\left\{\Phi^T \text{B}\mathbb{E}[\mathcal{E}\mathcal{E}^T](\text{B}\Phi - 2\text{B}\Phi H^{-1}\Phi^T \text{B}\Phi)\right\}, \tag{60}$$

$$= \text{Tr}\left\{\Phi^T \text{B}\Phi(I - 2H^{-1}\Phi^T \text{B}\Phi)\right\} \tag{61}$$

where we substituted in the definition of $\zeta$, then used that $\mathcal{E}$ and $\theta^0$ are independent, and that $\mathbb{E}[\theta^0] = 0$, and finally that $\mathbb{E}[\mathcal{E}\mathcal{E}^T] = \text{B}^{-1}$.

Writing $M = \Phi^T \text{B}\Phi$ and recalling that $H = (M + A)$, we have

$$\text{Tr}\,\Delta = \text{Tr}\left\{M(I - 2(M + A)^{-1}M)\right\}, \tag{62}$$

$$= \text{Tr}\left\{M(I - 2(A^{-1}M + I)^{-1}A^{-1}M)\right\}, \tag{63}$$

$$= \text{Tr}\left\{M(I - 2(I - (A^{-1}M + I)^{-1}))\right\}, \tag{64}$$

$$= \text{Tr}\left\{M(-I + 2(M + A)^{-1}A)\right\} \tag{65}$$

$$= -\text{Tr}\left\{M\right\} + 2\text{Tr}\left\{M(M + A)^{-1}A\right\}, \tag{66}$$

where we have used that $(A^{-1}M + I)^{-1}A^{-1}M = I - (A^{-1}M + I)^{-1}$ for the fourth equality. Now consider the isotropic prior case $A = \alpha I$ and recall the effective dimension is written as $\gamma = \text{Tr}\left\{M(M + A)^{-1}\right\}$. The above implies $\text{Tr}\,\Delta > 0$ if and only if $2\alpha\gamma > \text{Tr}\,\Phi^T \text{B}\Phi$.

### C.4 ANALYSING CONDITION AT CONVERGENCE

To gain some intuition for the condition at convergence, denote by $\lambda_1, \ldots, \lambda_{d'}$ the eigenvalues of $M$ (with multiplicity). We can use these to restate the condition as

$$2\alpha\gamma = 2\alpha \sum_{j=1}^{d'} \frac{\lambda_j}{\lambda_j + \alpha} > \sum_{j=1}^{d'} \lambda_j = \text{Tr}\{M\}. \tag{67}$$

This formulation of effective dimension gives an interpretation of a soft count of the number of dimensions for which $\lambda_j$ is larger than $\alpha$; in that sense, $\lambda_j$ measures how well determined the corresponding dimension of the weight vector $\theta$ is by the observed data. From here, note that

$$\frac{2\alpha\lambda_j}{\lambda_j + \alpha} > \min\{\lambda_j, \alpha\}, \tag{68}$$

and thus it is sufficient for $\text{Tr}\,\Delta > 0$ to hold at convergence that $\alpha > \lambda_j$ for all $j$ (but, of course, not necessary), yielding the intuition that $L'$ is preferred when the problem is heavily regularised.

## D DUAL FORM OF THE SAMPLE-THEN-OPTIMISE LOSS: MATHERON'S RULE

Both losses $L$ and $L'$ result in a random variable $\zeta \sim \Pi^0$ given by

$$\zeta = H^{-1}(\Phi^T \text{B}\mathcal{E} + A\theta^0). \tag{69}$$

Recalling that $H = A + \Phi^T \text{B}\Phi$ and using the push-through identity, we can express $\zeta$ equivalently as

$$\zeta = H^{-1}((H - \Phi^T \text{B}\Phi)\theta^0 + \Phi^T \text{B}\mathcal{E}) \tag{70}$$

$$= \theta^0 + H^{-1}\Phi^T \text{B}(\mathcal{E} - \Phi\theta^0) \tag{71}$$

$$= \theta^0 + A^{-1}(I + \Phi^T \text{B}\Phi A^{-1})^{-1}\Phi^T \text{B}(\mathcal{E} - \Phi\theta^0) \tag{72}$$

$$= \theta^0 + A^{-1}\Phi^T \text{B}(I + \Phi A^{-1}\Phi^T \text{B})^{-1}(\mathcal{E} - \Phi\theta^0) \tag{73}$$

$$= \theta^0 + A^{-1}\Phi^T(\text{B}^{-1} + \Phi A^{-1}\Phi^T)^{-1}(\mathcal{E} - \Phi\theta^0) \tag{74}$$

Now taking a sample of the posterior Gaussian process evaluated at input $x$ to be $G = \phi(x)\zeta$ and the corresponding sample of the prior process to be $G_0 = \phi(x)\theta^0$, premultiplying the above expression by $\phi(x)$ we obtain

$$G = G_0 + \phi(x)A^{-1}\Phi^T(\text{B}^{-1} + \Phi A^{-1}\Phi^T)^{-1}(\mathcal{E} - \Phi\theta^0) \tag{75}$$

which is Matheron's rule.

## E RESOLVING FEATURE SCALE INDETERMINACIES IN THE NN JACOBIAN WITH THE G-PRIOR

Antorán et al. (2022c) show that for NNs with normalisation layers, the Jacobian features $\phi(\cdot) = \nabla_w f(\bar{w}, \cdot)$ corresponding to each NN layer are scaled arbitrarily. To illustrate this, we divide the NN linearisation point into the concatenation of two weight vectors $\bar{w} = [\bar{w}_0, \bar{w}_1]$. We assume the layer containing $\bar{w}_0$ is followed by a normalisation layer, but not that containing $\bar{w}_1$, which leads to the invariance

$$f([k\bar{w}_0, \bar{w}_1], \cdot) = f([\bar{w}_0, \bar{w}_1], \cdot) \tag{76}$$

for all $k > 0$.

While $f$ is invariant to this scaling, the Jacobian feature embeddings $\phi(\cdot) = \nabla_w f(\bar{w}, \cdot)$ are not. We separate the embeddings as

$$\phi(x) = [\phi_0(\cdot), \phi_1(\cdot)] = [\nabla_{w_0} f(\bar{w}, \cdot), \nabla_{w_1} f(\bar{w}, \cdot)]. \tag{77}$$

Antorán et al. (2022c) show that, given a reference pair $([\bar{w}_0, \bar{w}_1], [\phi_0(x), \phi_1(x)])$, and for $\bar{w}_0$ normalised, scaling $\bar{w}_0$ by $k$ results in the pair $([k\bar{w}_0, \bar{w}_1], [k^{-1}\phi_0(x), \phi_1(x)])$. Thus, using a single prior precision parameter, the regularisation strength applied to the weights multiplying $\phi_0(x)$ relative

to those multiplying $\phi_1(x)$ will increase with $k$. The value of $k$, the scale of the linearisation point, depends on exogenous factors such as learning rate or batch size—and importantly is independent of the data, since it does not affect the output.

One way to resolve this is to assign the weights $\bar{w}_0$ and $\bar{w}_1$ separate regularisation parameters and learn these using the EM procedure outlined in Section 2. However, instead, consider using the g-prior normalised features $\phi'$ introduced in Section 4.3, and specifically, the scaling vector corresponding to normalised and non-normalised components $s = [s_0, s_1]$. For a reference pair $([\bar{w}_0, \bar{w}_1], [s_0, s_1])$ and for $\bar{w}_0$ normalised, the k-scaled pair is $([k\bar{w}_0, \bar{w}_1]$ and

$$[\mathrm{diag}(k^{-1}\Phi_0^T \mathrm{B}\Phi_0 k^{-1})^{\odot -\frac{1}{2}}, \ \mathrm{diag}(\Phi_1^T \mathrm{B}\Phi_1)^{\odot -\frac{1}{2}}] = [ks_0, s_1]$$

where $\odot$ denotes an elementwise power. Since the $k$-scaled features are $[k^{-1}\phi_0(\cdot), \phi_1(\cdot)]$, when applying the g-prior normalisation we recover a feature vector independent of $k$. This resolves the aforementioned pathology.

# F A PRACTICAL IMPLEMENTATION OF SAMPLE-BASED INFERENCE AND HYPERPARAMETER LEARNING FOR LINEARISED NEURAL NETWORKS

Algorithm 1 provides a high level overview of the procedure used for our experiments. This appendix expands on this, providing fully detailed algorithms for both sampled linearised Laplace applied to classification networks and the kernelised version of the method that we use for tomographic image reconstruction.

**Image classification**   Algorithm 2 provides an algorithm for linearised Laplace inference using the stochastic EM iteration presented in Section 3 for hyperparameter selection and the g-prior normalisation described in Section 4.3. Therein, $\mu$ denotes the softmax function. The curvature of the cross entropy loss at $x_i$, denoted $B_i$, is given by $B_i = \mathrm{diag}(p_i) - p_i p_i^T$ for $p_i = \mu(f(\bar{w}, x_i))$ our neural network's predictive probabilities. The notation $\odot$ refers to the elementwise product and to the elementwise power when used in an exponent. We refer to the Cholesky factorisation of a positive definite matrix as its $1/2$th power.

In order to limit computational cost, we sample the stochastic regularisation terms $(\theta_j^n)$, per (7), only once at the start. Not resampling these at each E step results in the optima of the sampling objective being close for successive iterations. This comes at the cost of a small bias in our estimator which we find to be negligible in practise. We separate $(\theta_j^n)$ into a sum consisting of a prior sample from $(\theta_j^0)$ and a data dependent term, denoted $(\theta_j')$. The former scales with $\alpha^{-1/2}$ while the latter with $\alpha^{-1}$ so this allows us to update each term in closed form each time $\alpha$ changes. We initialise our samples at $(\theta_j^0)$ at the first EM iteration. We warm-start the posterior mode $\bar{\theta}$ at the previous solution between iterations, initialising it to zero for the first iteration. We estimate the g-prior scaling vector $s$ by noting that it relates to $\theta_1'$ as $s = \alpha^{-1} \left(\mathbb{E}[\theta_1' \odot \theta_1']\right)^{\odot -1/2}$.

We optimise both our samples $\zeta$ and posterior mean $\bar{\theta}$ using stochastic gradient descent with a Nesterov momentum parameter of 0.9. We find that Polyak averaging is very effective at reducing gradient variance when optimising the sampling objective (per Dieuleveut et al., 2017). However, it has two limitations 1) it slows down optimisation, increasing the number of steps needed 2) it doubles the memory requirement needed to store posterior samples. This decreases the number of samples that can be optimised in parallel on a single hardware accelerator. Instead we employ a linear learning rate decay schedule, which we find to work nearly as well while not increasing computational burden. The regularised classification loss $\mathcal{L}$ is non-quadratic and thus Polyak averaging is no longer optimal (Bach, 2014). Thus here we also employ a linear learning rate decay schedule. For optimising both the sampling and classification loss objectives we find that gradient clipping helps prevent oscillations at the start of training and as a result speeds up convergence.

The key hyperparameters of our algorithm are the number of samples to draw for the EM iteration, the number of EM steps to run, and SGD hyperparameters: learning rate, linear decay rate, number of steps, batch-size and gradient clipping. Empirically, we find that at most 5 EM steps are necessary for hyperparameter convergence and that as little as 3 samples can be used for the algorithm without degrading performance. Choosing SGD hyperparameters is more complicated. However, we are aided by the fact that lower loss values correspond to more precise posterior mean and sample estimates.

---

**Algorithm 2:** Sampling-based linearised Laplace inference for image classification

---

**Inputs:** Linearised network $h$, linearisation point $\bar{w}$, observations $x_1, \ldots, x_n$, negative log-likelihood function $\ell$, initial precision $\alpha > 0$, number of samples $k$

**Function** *B(i)*:
$\quad\quad p_i \leftarrow \mu(h(\bar{w}, x_i))$
$\quad\quad$ **return** $\mathrm{diag}(p_i) - p_i p_i^T$

**for** $j = 1, \ldots, k$ **do**
$\quad\quad \theta_j^0 \sim \mathcal{N}(0, \alpha^{-1} I)$
$\quad\quad \theta_j' \leftarrow \alpha^{-1} \sum_{i=1}^n \phi(x_i)^T \varepsilon_j$ where $\varepsilon_j \sim \mathcal{N}(0, B(i))$
$\quad\quad \zeta_j \leftarrow \theta_j^0$
$\bar{\theta} \leftarrow 0$
$s \leftarrow \alpha^{-1} \left[ \frac{1}{k} \sum_{j=1}^k \theta_j'^{\odot 2} \right]^{\odot -1/2}$

**while** $\alpha$ *has not converged* **do**
$\quad\quad$ **for** $j = 1, \ldots, k$ **do**
$\quad\quad\quad\quad \zeta_j \leftarrow \mathrm{SGD}_z \left( \|\Phi(s \odot z)\|_B^2 + \alpha \|z - \theta_j^0 - (s \odot \theta_j')\|_2^2, \ \mathrm{init}=\zeta_j \right)$
$\quad\quad \bar{\theta} \leftarrow \mathrm{SGD}_\theta \left( \sum_{i=1}^n \ell(y_i, h((s \odot \theta), x_i) + \alpha \|\theta\|_2^2, \ \mathrm{init}=\bar{\theta} \right)$
$\quad\quad \hat{\gamma} \leftarrow \frac{1}{k} \sum_{j=1}^k \sum_{i=1}^n \|(\zeta_j \odot s)^T \phi(x_i)^T B(i)^{\frac{1}{2}}\|_2^2$
$\quad\quad \alpha' \leftarrow \hat{\gamma} / \|\bar{\theta}\|_2^2$
$\quad\quad$ **for** $j = 1, \ldots, k$ **do**
$\quad\quad\quad\quad \theta_j^0 \leftarrow \sqrt{\frac{\alpha}{\alpha'}} \theta_j^0$
$\quad\quad\quad\quad \theta_j' \leftarrow \frac{\alpha}{\alpha'} \theta_j'$
$\quad\quad \alpha \leftarrow \alpha'$

**Output:** Optimised precision $\alpha$ and weight samples $\zeta_1, \ldots, \zeta_k$

---

---

**Algorithm 3:** Kernelised sampling-based linearised NN inference for CT reconstruction

---

**Inputs:** Linearised network $h$, linearisation point $\bar{w}$, measurements $Y$, discrete Radon transform $U$, U-Net Jacobian $\Phi$, initial precision $\alpha > 0$, number of samples $k$, noise precision B

**Function** $\mathrm{Kvp}\,(v,\, \alpha,\, U\Phi,\, s,\, \mathrm{B}^{-1})$ **:**
$\quad\quad$ **return** $U\Phi(\alpha^{-1}\mathrm{diag}(s^{\odot 2}))\Phi^T U^T v + \mathrm{B}^{-1} v$

$s \leftarrow \left( \sum_{i<m} (U_i \Phi)^{\odot 2} \right)^{-1/2}$

**while** $\alpha$ *has not converged* **do**
$\quad\quad P \leftarrow \mathrm{Compute\text{-}preconditioner}(\mathrm{Kvp})$
$\quad\quad$ **for** $j = 1, \ldots, k$ **do**
$\quad\quad\quad\quad \zeta_j^0 \leftarrow U\Phi(s \odot \theta_j^0) + \mathcal{E}_j$ where $\mathcal{E}_j \sim \mathcal{N}(0, \mathrm{B}^{-1})$ and $\theta_j^0 \sim \mathcal{N}(0, A^{-1})$
$\quad\quad\quad\quad \mathrm{c}_j \leftarrow \mathrm{CG}\left( \mathrm{Kvp}, \zeta_j^0, \ \mathrm{precond.}=P \right)$
$\quad\quad\quad\quad \zeta_j \leftarrow \zeta_j^0 - U\Phi(\alpha^{-1}\mathrm{diag}(s^{\odot 2}))\Phi^T U^T \mathrm{c}_j$
$\quad\quad \delta \leftarrow U(\Phi\bar{w} - f(\bar{w}))$
$\quad\quad \mathrm{c} \leftarrow \mathrm{CG}\left( \mathrm{Kvp}, Y+\delta, \ \mathrm{precond.}=P \right)$
$\quad\quad \bar{\theta} \leftarrow s \odot \alpha^{-1}\Phi^T U^T \mathrm{c}$
$\quad\quad \hat{\gamma} \leftarrow \frac{1}{k} \sum_{j=1}^k \|U\Phi(s \odot \zeta_j)\|_2^2$
$\quad\quad \alpha' \leftarrow \hat{\gamma} / \|\bar{\theta}\|^2$
$\quad\quad \alpha \leftarrow \alpha'$

**Output:** Optimised precision $\alpha$

---

As a result, we can tune these parameters on the train data, no validation set is required. The specific hyperparameter values used in our experiments are provided in Appendix G.

A final thing to note is that due to the presence of normalisation layers and a dense final layer, for our classification networks, the constant-in-$\theta$ terms cancel in the linearised model and we are left with $h(\theta, x) = \phi(x)\theta$ (Antorán et al., 2022c). In our algorithm, this fact is only relevant to the computation of the posterior mode $\bar{\theta}$ as the optima of $\mathcal{L}(h(\theta, \cdot))$.

**Tomographic reconstruction**   Algorithm 3 is the kernelised version of algorithm 2 that we use for tomographic reconstruction. This problem is described in detail in Appendix G.3.

Distinctly from the image classification setting, tomographic reconstruction is a regression problem for which we use a Gaussian likelihood with fixed noise precision $\mathrm{B} = I$. The linear model's loss function $\mathcal{L}$ is thus quadratic and the Laplace approximation is not needed. Both the sample loss and the linear model's loss can be optimised in closed form by solving a linear system of equations given by the observation covariance, i.e. the kernel matrix, $U\Phi(\alpha^{-1}\mathrm{diag}(s^{\odot 2}))\Phi^T U^T + \mathrm{B}^{-1}$ where the linear operator $U$ represents the discrete Radon transform and combines with the U-Net Jacobian to build the feature embedding $U\Phi$.

We solve against the kernel matrix using the preconditioned conjugate gradient (CG) method described by Gardner et al. (2018). As a preconditioner, we compute a 400-dimensional randomised eigendecomposition (alg. 5.6 in Halko et al., 2011) preconditioner, which we invert using the Woodbury identity. We find the preconditioner to provide important speedups to CG convergence and we re-estimate it after every hyperparameter update. Both computing the preconditioner and running preconditioned CG optimisation only interact with the kernel matrix by computing its products with vectors. Our algorithm defines our kernel vector product `Kvp` routine explicitly, as it is central to our implementation. We find that the GPyTorch CG implementation does not benefit from warm-starting the solution vector. Consequently, we re-draw prior and noise samples $(\theta^0, \mathcal{E})$ at every E-step.

Similarly to image classification, the key hyperparameters are the number of samples to draw for the EM iteration, the number of EM steps to run, and CG optimisation hyperparameters. Again, the number of samples can be kept low (e.g. 2) and we find around 5 steps to suffice for convergence of the prior precision $\alpha$. The key conjugate gradients hyperparameters are the tolerance at which to stop optimisation and the maximum number of optimisation steps if the tolerance is not reached. We provide our choices in Appendix G.3 but note that our use of a large preconditioner results in CG always hitting the desired low error tolerance within 10 steps and never stopping due to reaching the maximum number of steps. In turn, this makes our kernelised EM algorithm notably faster than its primal form SGD-based counterpart.

A particularity of this setting is that the U-Net does not have a dense final layer. As a result, the constant-in-$\theta$ terms in the linearised function $h$ do not cancel (see Section 4.1), leading to the appearance of the target offset term $\delta$ when solving for the posterior mean.

## G   EXPERIMENTAL DETAILS

In this appendix we provide experimental details and hyperparameter settings omitted from the main text.

### G.1   MNIST EXPERIMENTS

MNIST $m=10$ way classification experiments were performed using the LeNet-style CNN architectures of increasing size employed by Antorán et al. (2022c): "LeNetSmall" ($d'=14634$), "LeNet" ($d'=29226$) and "LeNetBig" ($d'=46024$). We note that these models contain batch normalisation layers. Each model was trained with using SGD with momentum of 0.9 for 90 epochs with a learning rate drop of a factor of 10 every 30 epochs. The MNIST dataset was downloaded from `PyTorch torchvision`. We employ standard per-channel mean and std-dev standardisation preprocessing and two pixel shift and crop data augmentation. For posterior mode optimisation and sampling, we do not perform data augmentation as to avoid cold posterior effects (Izmailov et al., 2021). The details of our SGD approaches to convex optimisation for obtaining posterior modes and samples are as follows

- **Posterior mode optimisation**: The linearised NN weights are trained using SGD with a Nesterov momentum coefficient of 0.9, and batch size 1000 for 40 epochs. We clip gradients to a maximum norm of 1. We use an initial learning rate of $1e-2$ when using standard isotropic or layerwise Gaussian priors, and 1 for the g-prior. We employ a linear decay schedule that reduces the lr by a factor of 330 over the first 75% of the training procedure and holds it constant afterwards.

- **Sampling**: We optimise 32 samples in parallel using SGD with Nesterov momentum ($=0.9$) and a batch size of 1000 for 20 epochs. For standard Gaussian priors (isotropic and layerwise), we use a learning rate of $2e-1$, whereas for the g-prior, we find a higher learning rate of 200 to work best.

**Hyperparameter optimisation**: We tuned the learning rate, decay schedule and gradient clipping strength using a rough grid search over multiple orders of magnitude. We chose the settings that reached the lowest loss values. These can be evaluated with just the train set. We chose the largest batch size that could accommodate optimising 32 samples in parallel on a single hardware accelerator. We note that posterior mode and sample optimisation converge in less than half of the total epochs we use for their optimisation. The numbers of epochs chosen were set to be large enough to ensure convergence and not tuned. A decrease in computational cost can likely be achieved by stopping sample optimisation earlier.

**Baseline methods.** For the comparison of learning a single precision hyperparameter and layerwise hyperparameters in Figure 3, we extend the M-step update to as $\alpha_l = \gamma_l/\|\bar{\theta}_l\|_2^2$ where $l$ indexes each layer's attributes, as done in (Mackay, 1992; Tipping, 2001). For the MAP, diagonal covariance and KFAC covariance baselines, we use the same pre-trained models when possible (i.e. not for the ensembles or dropout baselines). Since all baselines share the same linearisation point, they also share the same mean predictions. Differences in performance among baselines are thus only due to differences in uncertainty estimation. The diagonal approximation to the covariance is constructed by first computing the diagonal of the Hessian $M$ and the inverting it. For the KFAC covariance approximation, we exploit the equivalency between the Generalised Gauss Newton matrix (i.e. the Hessian of the linear model $h$) and the Fisher information matrix for exponential family likelihoods (i.e. the categorical). This allows us to formulate the Hessian as an expectation of likelihood gradients, which in turn we approximate using a single sample per training observation, as in (Daxberger et al., 2021a). For completeness, we also state the probit approximation for sampled predictive posteriors over logits. For input $x$ and samples $\zeta_i, \ldots, \zeta_k$, the predictive probability for class $i \in \|1, \ldots, m\|$ is given by

$$\text{softmax}\left( f(\bar{w}, x) \odot (1 + \frac{\pi}{2k} \sum_{j<k} (\phi(x)\zeta_j)^{\odot 2})^{\odot -0.5} \right)_i .$$

## G.2 CIFAR100 CLASSIFICATION

CIFAR100 $m=100$ way classification experiments were performed using ResNet18 models ($d' \approx 11M$) with specific architecture details matching the `PyTorch torchvision implementation`. We train these models using SGD with momentum of 0.9 for 300 epochs. The starting lr is 0.1 and we reduce it by a factor of 10 every 100 epochs. The CIFAR100 dataset was also downloaded using `torchvision` and our data preprocessing and augmentation also follow the default implementation from this library. For posterior mode optimisation and sampling, we do not perform data augmentation. The SGD details used to solve the convex optimisation problems required for obtaining posterior modes and drawing samples are as follows

- **Posterior mode optimisation**: The linearised NN weights are trained using SGD with Nesterov momentum ($=0.9$) and a batch size of 2000 for 40 epochs. We employ a linear decay learning rate schedule with an initial learning rate of $1e-1$. It is decreased by a factor of 330 over the first 75% of training, and then held constant. We also employ gradient clipping with maximum norm$=0.1$.

- **Sampling**: We optimise 6 samples in parallel using SGD with Nesterov momentum ($=0.9$) and a batch size 100 for 20 epochs. All other details match those of posterior mode optimisation.

Upon convergence of the EM algorithm, we draw 64 further samples using the optimal prior precision by following the optimisation procedure described above. We initialise these samples at prior samples drawn with the optimised prior precision.

**Hyperparameter optimisation**: We tuned the learning rate, decay rate and gradient clipping strength using a rough grid search over orders of magnitude. We also chose the largest batch size that for which we could simultaneously optimise 6 samples in parallel on a single hardware accelerator. Similarly to the MNIST experiments, we did not optimise the number of optimisation epochs and instead chose large values that would ensure convergence. It is likely that our EM iteration can be sped up by decreasing the duration of the convex optimisation routines.

Details for baselines and hyperparameters not mentioned explicitly in this subsection match those given for MNIST in the previous subsection.

### G.2.1 EFFICIENT $\kappa$-ADIC SAMPLING

Osband et al. (2022; 2021) introduced *dyadic* test input sampling ($\kappa = 2$) as a practical way of evaluating joint predictions in discriminative tasks. This method samples $\kappa = 2$ random anchor points from the test dataset, and then randomly resamples them to create a batch of size $\tau = 10$. Test log-likelihood is evaluated jointly for each batch as

$$\log \int \exp \left( \sum_{i \leq \tau} \ell(y_i, f(\theta, x_i)) \right) d\Pi,$$

for $f$ the model being evaluated and $\Pi$ its posterior distribution over model parameters. This quantity can be estimated with posterior samples $\zeta_1, \ldots, \zeta_k \sim \Pi$ as

$$\log \frac{1}{k} \sum_{j \leq k} \exp \left( \sum_{i \leq \tau} \ell(y_i, f(\zeta_j, x_i)) \right).$$

We extend this evaluation approach to larger $\kappa$ and $\tau$ values without increasing computational cost. We randomly sample $\kappa$ integers $\{b_1, \ldots, b_\kappa\}$ such that they sum to $\tau$, i.e $\sum_i^\kappa k_i = \tau$. The joint log-likelihood over the batch of size $\tau$ with $\kappa$ unique datapoints can then be estimated as

$$\log \frac{1}{k} \sum_{j \leq k} \exp \left( \sum_{l \leq \kappa} b_l \ell(y_l, f(\zeta_j, x_l)) \right).$$

where the inner sum is over the $\kappa$ distinct elements in the batch instead of the "total batch size" $\tau$. This is equivalent to the formulation proposed in Osband et al. (2022) for dyadic sampling, when $\kappa = 2$ and $\tau = 10$. We note that it is not possible to achieve *augmented dyadic* sampling, as described in (Osband et al., 2021), with this approach. However the authors mention that there is not a significant difference in the relative performance of methods when using augmented dyadic sampling compared to regular dyadic sampling. We introduce a final step however, which is to repeat the computation for multiple shuffles (10) of the test dataset. This eliminates variance in our results from the choice of the $\kappa$ observations which get grouped together in each batch.

### G.3 TOMOGRAPHIC RECONSTRUCTION

**Problem setup** Tomographic reconstruction consists in solving a linear inverse problem in imaging where we observe a set of measurements $y \in \mathbb{R}^m$, which we assume to be generated as $y = Ux^* + \eta$ for $U \in \mathbb{R}^{m \times d}$ the discrete Radon transform, $x^* \in \mathbb{R}^d$ the image to reconstruct and $\eta \sim \mathcal{N}(0, I)$ random noise. We have $m \ll d$, making the problem underconstrained. Our specific tomographic reconstruction task closely follows the one from Barbano et al. (2021). We perform a sparse-view reconstruction of an image of a slice of a walnut from a sub-sampled set of measurements. Specifically, from the full measurement set of (Der Sarkissian et al., 2019a), which containing scans at 1200 equidistant angles over $[0, 360°)$, we choose our measurement set by subsampling angles by a factor of either 10x or 20x, leading to measurements of size $m = 15360$ or $m = 7680$. As in Barbano et al. (2021); Antorán et al. (2022b); Barbano et al. (2023), we reduce the original 3D scan geometry to the 2D slice of interest by selecting the relevant subset of measurement pixels. We assemble the Radon operator $U$ as a sparse matrix taking in an image of resolution $(501\text{px})^2$ and outputting a measurement tensor coherent with the described geometry.

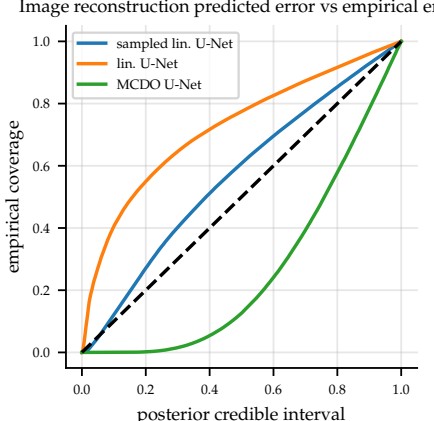 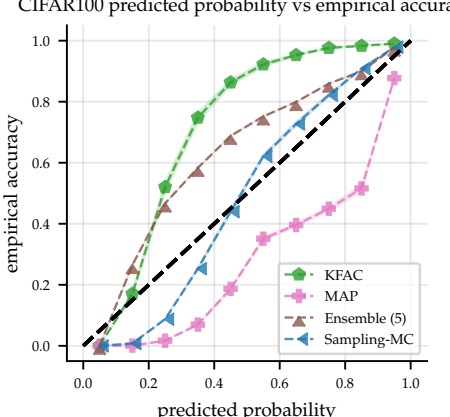

Figure 8: Left: empirical coverage of test targets for posterior credible intervals of increasing width for our U-net tomographic reconstruction experiment (Section 5.3). Right: confidence vs accuracy plot (also known as a reliability diagram) for our CIFAR100 classification experiment (Section 5.2).

**Methods** To provide a reconstruction, we use the Deep Image prior (Ulyanov et al., 2020) which trains the parameters $w \in \mathbb{R}^{d'}$ of a fully convolutional U-Net autoencoder $f : \mathbb{R}^{d'} \to \mathbb{R}^d$, where the input is fixed and thus absent from our notation, until a satisfactory reconstruction $f(\bar{w})$ is obtained. The U-Net network architecture is the one proposed in (Baguer et al., 2020). The optimisation of the U-Net parameters follows Barbano et al. (2021), although we note that faster optimisation strategies exist (Barbano et al., 2023). To estimate the uncertainty in this reconstruction, we linearise the U-Net around $\bar{w}$, as described in Section 4.1. This leaves us with a model affine in the parameters and with design matrix $U\Phi \in \mathbb{R}^{m \times d'}$. We may now proceed with linear model inference. While (Antorán et al., 2022b) use the traditional EM iteration described in Section 2, we use the sample-based one from Section 3. For evaluation, we use the non-sparse reconstruction (using 1200 angles) provided by (Der Sarkissian et al., 2019a) as the ground truth image $x^*$. To evaluate joint log-likelihoods we estimate the predictive covariance matrix for patches of neighbouring pixels using samples. Covariance matrix estimates from samples are known to be unreliable. We use the stabilised formulation of (Maddox et al., 2019): $\hat{\Sigma} = \frac{1}{2k} \left[ \sum_{j=1}^{k} \hat{x}_j^2 + \hat{x}_j \hat{x}_j^T \right]$ for $(\hat{x}_j)_{j=1}^{k}$ samples from the predictive posterior over a patch. We then construct predictive distributions over pixels as $\mathcal{N}(f(\bar{w}), \hat{\Sigma})$.

**Hyperparameters** We employ a low CG tolerance of $1e-3$ and a maximum number of iterations of $150$, which is never reached in practise as the error tolerance level is always hit in less steps.

## H CALIBRATION OF PREDICTIVE DISTRIBUTIONS

This appendix evaluates the calibration of the predictive distributions provided by the methods under consideration in our CIFAR100 classification experiment (Section 5.2) and U-net image reconstruction experiment (Section 5.3).

For classification, we separate our predicted probabilities into 10 equal width bins between 0 and 1. For each bin, we plot the proportion of targets that coincide with the class for which the predicted probability falls into the bin. This is shown on the right hand side of Figure 8. Consistent with the results from the main text, KFAC overestimates uncertainty at all confidence levels whereas MAP underestimates it. Both sample-based linearised Laplace and ensembling show significantly improved calibration. While ensembles show a small amount of uncertainty overestimation consistently, our method underestimates uncertainty for low predicted probabilities and overestimates it for large predicted probabilities.

For image reconstruction regression, we first compute normalised residuals by subtracting our predictions from the targets and dividing by the predictive standard deviation. Our predictive distribution for these normalised residuals is the centered unit variance Gaussian. We consider

posterior credible intervals centered at 0 and of increasing width and plot the proportion of test points that fall within them in the left side plot of Figure 8. We find dropout inference to underestimate the magnitude of the residuals across all credible interval widths. Linearised inference with a single EM step, as in (Antorán et al., 2022b), consistently overestimates uncertainty. Our approach, which performs 5 steps of EM, overestimates uncertainty, but to a much smaller degree, presenting the best overall calibration. The latter two approaches consist of the same model but with different regularisation strength. The difference between the two reveals the paramount importance of tuning the prior precision hyperparameter well.

# I  ADDITIONAL EXPERIMENTS

This appendix contains additional experiments and baselines that supplement the experimental results provided in the main text.

## I.1  COMPARING PRIMAL AND DUAL EFFECTIVE DIMENSIONALITY ESTIMATORS

Our main-text experiments employ the kernelised effective dimension estimator introduced in (5). A different unbiased estimator may be obtained in primal form following the derivation provided in Appendix B.1. Figure 9 compares both estimators when applied to the 1d toy problem used to generate Figure 1 from the main text. In particular, we use a 2 hidden layer MLP with layernorm after every hidden layer and the "Matern" dataset of Antorán et al. (2020). We use 8 samples from the exact linearised Laplace posterior to compute effective dimension estimates and repeat this procedure 1000 times to characterise the behaviour of each estimator. As a reference, we also compute the exact effective dimension using eigendecomposition.

Both estimators present distributions centered at the true effective dimension value. However, the prediction space (kernelised) estimator presents a much lower variance of 9.16 as opposed to 654.19 from the weight space estimator. Additionally, the weight space estimator distribution places a substatial amount of probability mass on negative effective dimension values. From the form of (19), we see that this is due to our 8-sample estimator overestimating posterior variance. On the other hand, the kernelised estimator in (5) can only produce positive values.

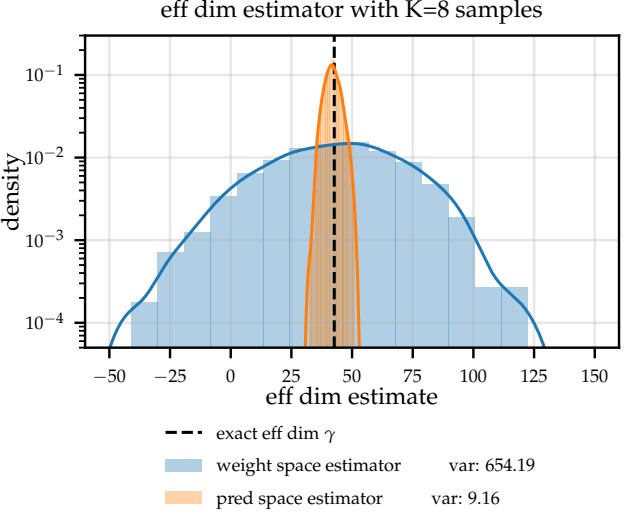

Figure 9: Histogram, with bin heights normalised to represent density estimates, of the effective dimension estimates produced by the primal form (weight space) estimator (19) and the kernelised (prediction space) estimator (5). Both distributions are roughly centered at the true effective dimension but the kernelised estimator presents much lower variance.

### I.2 EVALUATING APPROACHES TO UPDATING HYPERPARAMETERS IN THE M-STEP

This section empirically motivates the fixed-point iteration M-step introduced by Mackay (1992), and described in Section 3, by comparing it with alternative approaches to updating hyperparameters. In particular, we compare Mackay's update with the standard Laplace M-step evidence, denoted $\mathcal{M}$ and given in (3) and Appendix B.2, and a Gaussian ELBO with optimised mean and covariance. The latter two objectives differ in that the regulariser appears inside of the log-determinant term in $\mathcal{M}$, while the ELBO's covariance does not change with the regulariser while performing the M-step. Both of these objectives differ from the Mackay update in that they provide an objective which requires gradient-based optimisation in the M-step. Instead, the Mackay update has a closed-form.

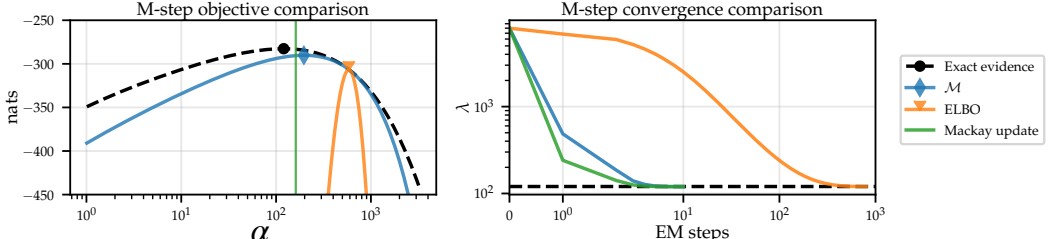

Figure 10: Left: exact linear model evidence for a linearised 2 hidden layer MLP with layer normalisation together with the lower bound presented in (3), $\mathcal{M}$, and an ELBO where the Gaussian posterior covariance is decoupled from the regulariser. All curves use an initial regulariser of $\alpha = 500$ and have a marker placed at their optima. Right: values of the regularisation strength $\alpha$ obtained at successive EM iterations while using the different update strategies under consideration for the M step. Note that when we assume access to the exact evidence function, the regulariser converges in a single step and no EM iteration is necessary.

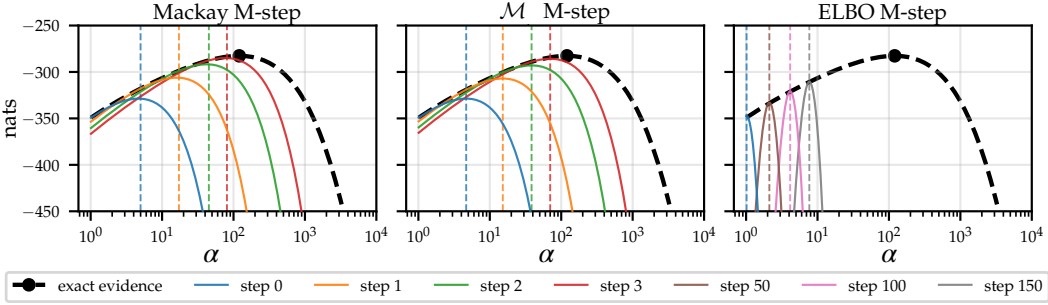

Figure 11: Exact linear model evidence for a linearised 2 hidden layer MLP with layer normalisation together with the lower bound presented in (3), $\mathcal{M}$ (left and middle plots), and an ELBO, where the Gaussian posterior covariance is decoupled from the regulariser (right hand side plot), at different EM steps. We update the regularisation strength with Mackay's fixed point iteration for the left side plot. Note that $\mathcal{M}$ curves are shown in this plot. We maximise $\mathcal{M}$ in the middle plot and we maximise the ELBO in the right hand side plot. All curves use an initial regulariser of $\alpha = 5$ and we place a vertical dashed line at each step's update.

The plot on the left of Figure 10 compares the exact linearised Laplace evidence for a 2 hidden layer MLP with layernorm trained on the toy dataset of Antorán et al. (2020) with the bound $\mathcal{M}$ (3) and the decoupled ELBO. The initial regulariser is set to $\alpha = 500$. The ELBO is only tight for regulariser values very close to initialisation, resulting in very small M steps. $\mathcal{M}$ is tangent to the evidence at the same point as the ELBO but presents a much better approximation as we move away from $\alpha = 500$. The optimum of $\mathcal{M}$ is much closer to the optimum of the evidence. The Mackay update does not use a lower bound but instead provides an updated value for $\alpha$ which is even closer to the optimum of the evidence. The right hand side plot shows the change in the regularisation parameter across successive M-steps using the update methods under consideration. The Mackay M-step converges to the optima of the evidence in 2 steps. Using $\mathcal{M}$ as an objective results in convergence after 5

steps. On the other hand, the ELBO update requires around 100 steps. Figure 11 further illustrates hyperparameter learning in the 1d toy setting by showing the successive lower bounds obtained by each of the approaches under consideration at each M-step. Interestingly, the Mackay update produces regulariser updates that almost exactly maximise $\mathcal{M}$.

Figure 12 compares the evidence lower bounds of the form of $\mathcal{M}$, given in (3), when using different covariance matrix approximations in the MNIST classification setup presented in Section 5.1. In particular, we consider the full-covariance Laplace evidence, which we note does not match the exact model evidence due to the non-quadratic classification loss, the KFAC approximation to the covariance (labelled KFAC GGN), a single-sample KFAC Fisher estimate of the covariance, the KFAC empirical Fisher matrix (Immer et al., 2021a), and a diagonal Laplace covariance. We also include a 16 sample estimate of the ELBO described above. In all cases, we initialise the regulariser at an optima found by applying the EM algorithm while using the full covariance $\mathcal{M}$ in the M-step. In this way, we may use the deviation of different objectives' optima from the optima of $\mathcal{M}$ as estimates of the bias in their corresponding approximations.

Figure 12 shows the KFAC and KFAC-Fisher approximations result in a systematic overestimation of the evidence optima which grows with model size. This issue is even more pronounced for the diagonal covariance approximation. On the other hand, we find the empirical Fisher to provide an accurate approximation. A similar finding is reported by (Immer et al., 2021a). This is surprising, given that the empirical Fisher is known to provide a heavily biased estimate of loss curvature and thus perform poorly for optimisation tasks (Kunstner et al., 2019). The sample-based ELBO shows close to no bias when using 16 samples. This result agrees well with our experiments from Section 5.2, where the sample-based EM algorithm behaves well even when using very few samples.

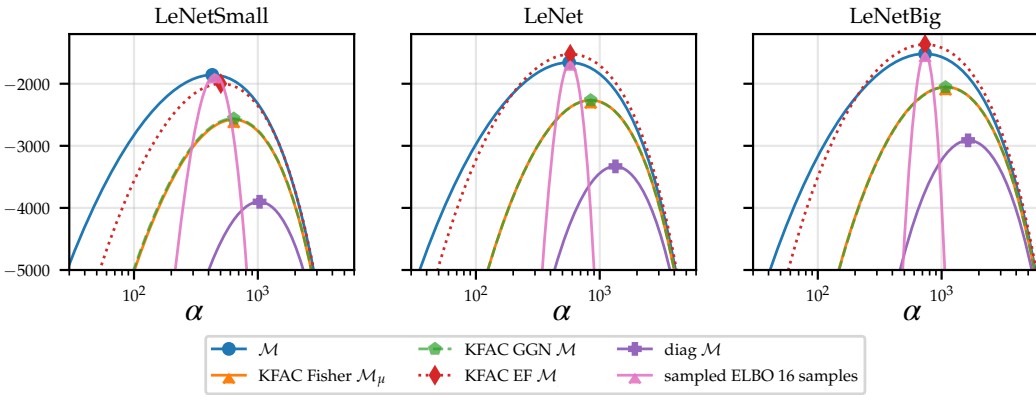

Figure 12: Full covariance linearised Laplace evidence $\mathcal{M}$ together with approximations to this curve that rely on different covariance matrix approximations. We consider convolutional networks of increasing size (left to right) trained on the MNIST dataset.

### I.3 CIFAR100 CLASSIFICATION

**Additional Baselines** . In the main text, we report the test log-likelihood obtained by our method as well as that of a point-estimated NN (MAP), an ensemble of 5 of point-estimate NNs (Ensemble 5), and linearised Laplace with a KFAC-approximated posterior covariance matrix (KFAC). Here, we report further comparisons with other baselines standard-in-literature: a diagonal approximation of the Laplace covariance matrix (diag), a Laplace approximation over a selected subset of the full NN weight space (subnetwork*) (Daxberger et al., 2021b), and a Laplace approximation over only the last-layer weights of the NN with a KFAC covariance matrix approximation (KFAC-LL*) (Eschenhagen et al., 2021). Note that the last layer contains 51200 weights and thus its full Laplace covariance matrix is too large to invert on our A100 GPUs. We distinguish the last two methods with a star (*) to denote that they require cross-validation with a held-out set to tune the regularisation strength hyperparameter. In particular, we use 50 held-out points from the test set, and evaluate these methods on the remaining 9950 points. This gives these methods a slight advantage over the approaches to uncertainty quantification considered in the main text. Similarly to the main

text, we estimate the KFAC and sampling posterior distributions with 64 samples. We use exact marginalisation, computing full Jacobians, for the diagonal covariance approximation.

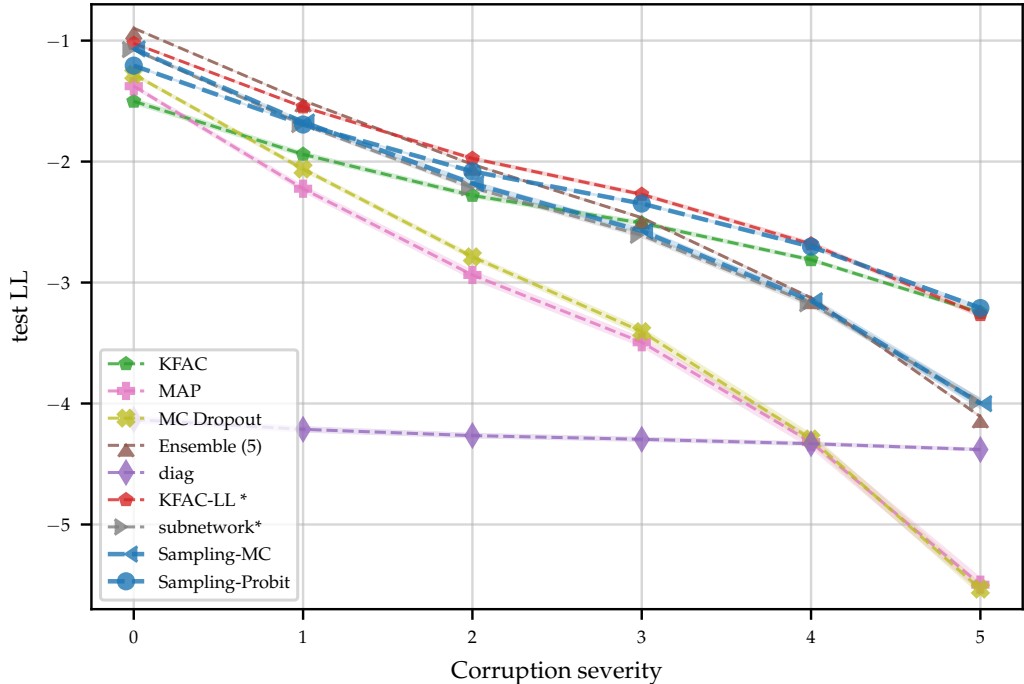

Figure 13: Performance under distribution shift for additional inference baselines applied to ResNet18 on the CIFAR100 dataset. We note that KFAC-LL and subnetwork inference require a held-out validation set to tune hyperparameters and thus we mark them with a star (*).

**Robustness to distribution shift** We provide the test log-likelihood results obtained with all methods under data corruption of increasing intensity in Figure 13. Following the standard setup in the literature (Daxberger et al., 2021b;a; Eschenhagen et al., 2021), we employ the multi-class probit approximation to map Gaussian posteriors over NN outputs to class probabilities for all methods except ensembles and MAP. However, when combined with our sampling approach, we find the probit approximation to overestimate uncertainty in-distribution. We illustrate this by plotting an additional curve for sample-based inference with Monte-Carlo marginalisation of the Gaussian distribution over NN outputs. This approach provides stronger in-distribution performance which comes very close to that of ensembles, subnetwork inference (*), and KFAC-LL (*). The strong performance of the latter two methods reveals the relevance of selecting a good regularisation strength parameter to uncertainty quantification with Laplace-style methods. In the out-of distribution setting, the probit's increased uncertainty results in larger log-likelihood scores than Monte Carlo Marginalisation. KFAC-LL performs very competitively both with ensembles in-distribution and with our approach in the out-of-distribution setting.

**Joint LL**. We report marginal and joint test log-likelihood for the KFAC-LL and subnetwork inference baselines in Table 3. We use the same $\kappa$-adic sampling setup as in Section 5.2, marginalising the Gaussian posterior over network outputs with Monte Carlo for all methods. KFAC-LL is once again quite competitive with our approach in terms of both marginal and joint LL.

**Predictive uncertainty vs number of samples**. In the main text, we report our method's predictive performance when drawing 64 samples. In Figure 14, we plot the degradation in test log-likelihood for the standard and progressively corrupted CIFAR100 test sets when decreasing the number of samples used for prediction. We provide results for both Monte Carlo and probit marginalisation. Our results, show two main trends: 1. Monte Carlo marginalisation provides better results in-distribution for all numbers of samples. This is coherent with our above observation that the probit approximation results in uncertainty overestimation. On the other hand, Monte Carlo marginalisation is unbiased. 2. The probit approximation benefits less from increased numbers of samples. This is expected, since

Table 3: Comparison of methods' marginal and joint prediction performance for ResNet18 on CIFAR100. We include baselines that require validation-based tuning of the regularisation strength, marking them with a star (*).

| | $\kappa$ | MAP | Ensemble (5) | KFAC | Sampling | KFAC-LL * | subnetwork * |
|---|---|---|---|---|---|---|---|
| marginal LL | 1 | $-1.40 \pm 0.00$ | $\mathbf{-0.90 \pm 0.00}$ | $-1.12 \pm 0.01$ | $-1.07 \pm 0.01$ | $-1.06 \pm 0.01$ | $-1.21 \pm 0.01$ |
| | 2 | $-13.97 \pm 0.01$ | $-6.86 \pm 0.01$ | $\mathbf{-4.92 \pm 0.04}$ | $-5.14 \pm 0.04$ | $-5.41 \pm 0.05$ | $-8.38 \pm 0.07$ |
| joint LL | 3 | $-27.89 \pm 0.03$ | $-14.17 \pm 0.03$ | $-10.83 \pm 0.12$ | $\mathbf{-10.77 \pm 0.09}$ | $-11.15 \pm 0.12$ | $-16.59 \pm 0.13$ |
| | 4 | $-41.83 \pm 0.03$ | $-22.29 \pm 0.04$ | $-19.02 \pm 0.22$ | $\mathbf{-18.04 \pm 0.18}$ | $-18.21 \pm 0.18$ | $-25.47 \pm 0.18$ |
| | 5 | $-55.89 \pm 0.02$ | $-31.07 \pm 0.09$ | $-29.40 \pm 0.40$ | $\mathbf{-26.75 \pm 0.26}$ | $-26.50 \pm 0.26$ | $-34.91 \pm 0.30$ |

the probit method discards covariances in the distribution over network outputs. The probit predictive distribution has less degrees of freedom and can thus be estimated better with less samples.

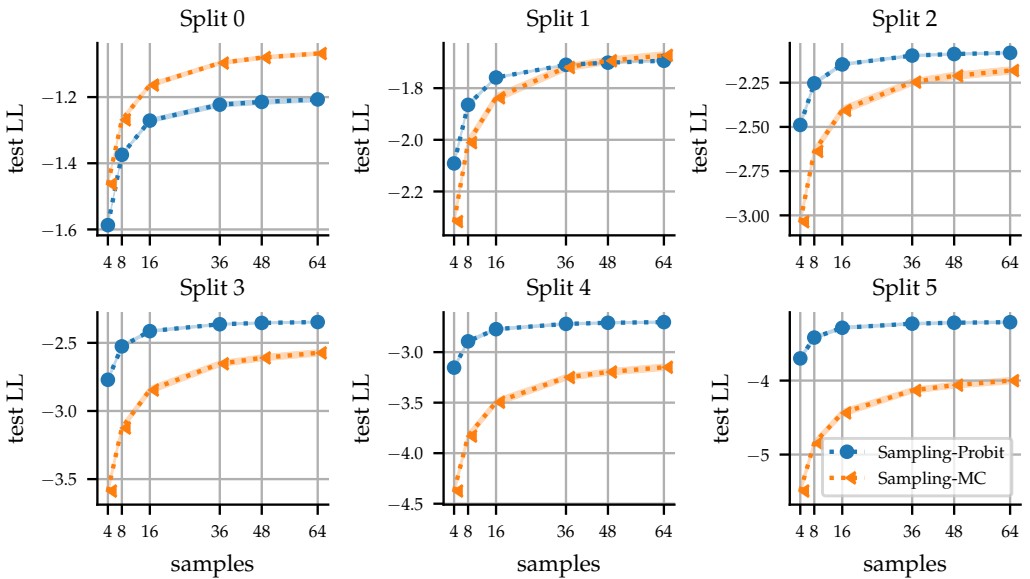

Figure 14: Predictive Performance for sample-based inference on the CIFAR100 in-distribution and corrupted test sets while varying the number of samples used for estimating the predictive distribution.

### I.4  TOMOGRAPHIC RECONSTRUCTION

**Stability of stochastic EM in the** $m = 15360$ **setting**    Figure 15 shows the prior precision values, effective dimension values and test log-likelihood values obtained at each EM step for the larger 120 angle ($m = 15360$) image reconstruction task. The regularisation strength converges faster in this more data-rich setting than in the 60 angle setting considered in the main text (Figure 6 ), with convergence occurring after 1 EM step instead of 2. We see a slightly larger sensitivity to the number of samples in this larger setting. However, the difference in test LL obtained after running stochastic EM with 2 and 256 samples remains smaller than 0.01 nats.

**Further analysis of uncertainty calibration**    The rightmost plot in Figure 15 compares the joint test log-likelihood obtained by our method and MC dropout on image patches of increasing size when the observation dimension is set to $m = 15360$. Similarly to the results shown in the main text, our method performs better across all patch sizes. Qualitatively, Figure 16 shows the sample-based approach to yield uncertainty estimates with a much larger dynamic range; some pixel regions are assigned large errorbars, while others are assigned small errorbars. MCDO produces less fine-grained outputs and assigns relatively small errorbars to whole sections of the image.

Finally, Figure 17 and Figure 18, compare the reconstruction error and uncertainty histograms for both uncertainty quantification methods under consideration for both the $m = 7680$ and $m = 15360$ settings. In both plots, sample-based linearised Laplace inference slightly overestimates uncertainty.

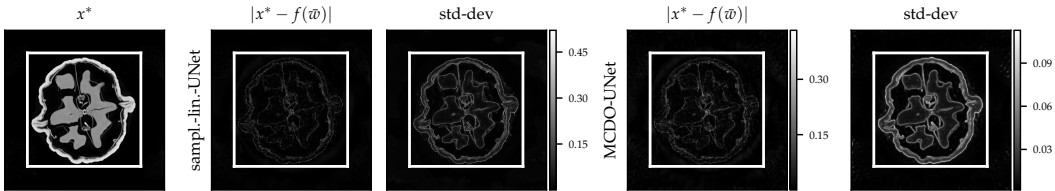

Figure 15: Sample-based EM iteration convergence for tomographic reconstruction given $m = 15360$. The prior precision $\alpha$ (left), the effective dimension $\hat{\gamma}$ (middle left), and the marginal test log-likelihood (LL) (middle right) are plotted as a function of the EM step. The plot on the right shows the joint test LL across image patches of increasing size for sampling inference and an MC dropout baseline (MCDO).

Figure 16: Original $501 \times 501$ pixel walnut image and reconstruction error for a $m{=}15360$ dimensional observation, along with pixel-wise std-dev obtained with sampling lin. Laplace and MCDO.

MCDO systematically underestimates uncertainty for the pixels where the reconstruction error is largest. Interestingly, our method shows to be slightly worsely calibrated in the more data-rich setting, as the reconstruction error decreases faster than the predictive standard deviation.

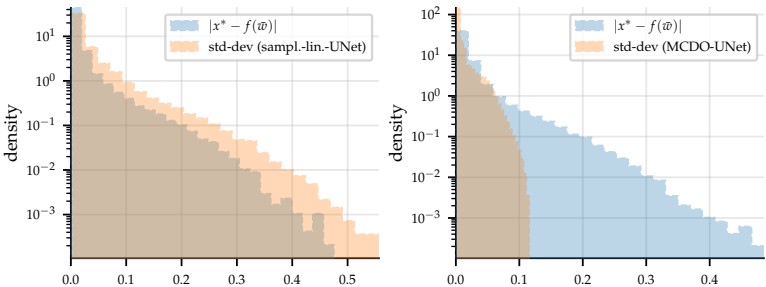

Figure 17: Histogram of the absolute pixelwise error computed between the reconstructed walnut image given $m = 7680$ observations and the ground-truth for both lin.-UNet and MCDO-UNet. We also include the histograms of both methods' predictive standard deviations across pixels.

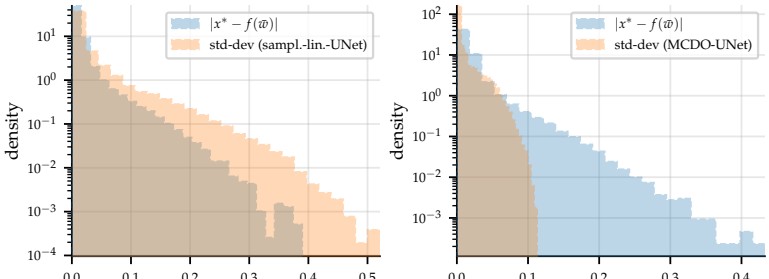

Figure 18: Histogram of the absolute pixelwise error computed between the reconstructed walnut image given $m = 15360$ observations and the ground-truth for both lin.-UNet and MCDO-UNet. We also include the histograms of both methods' predictive standard deviations across pixels.

