# OpenReview forum: "Sampling-based inference for large linear models, with application to linearised Laplace"
_ICLR.cc/2023/Conference — ICLR 2023 poster_

### Official Review · Reviewer_ZeYJ · 2022-10-19

**Confidence:** 4
**Correctness:** 4
**Technical Novelty And Significance:** 3
**Empirical Novelty And Significance:** 3
**Recommendation:** 8

**Clarity, Quality, Novelty And Reproducibility:**

The paper is written in a clear and comprehensible manner. The proposed approach combines several known approaches into a joint new one and provides a novel solution to the task. The discussion in the appendix together with the provided code seems to provide full reproducibility.


**Strength And Weaknesses:**

The authors tackle an important task as the linearised Laplace is a popular approach, especially in the Bayesian Deep Learning literature. The experimental evaluation is extensive and considers both small/large scale classification tasks as well as an image reconstruction task. A minor weakness is the lack of downstream evaluation of the quality provided by the method's predictive uncertainty. However, that can be justified by the necessary limit of the breadth of the work given the page constraints.


**Summary Of The Paper:**

The authors consider the task of improving the scalability of the popular linearised Laplace method for which they develop a sampling-based EM approach. The value of the proposed approach is then demonstrated in several experimental settings.


**Summary Of The Review:**

The paper sets out to solve a well-defined task and delivers both concerning the theoretical as well as empirical side.

---

> ### Author Response · Authors · 2022-11-10
> **Thank you for the review and comments!**
>
> We thank the reviewer for their time reading our paper and for their kind words.
>
> **Downstream evaluation**
>
> A downstream application in the style of a bandit or reinforcement algorithm using our method would indeed be interesting, particularly the E-step. We considered doing this, but we were worried that introducing the relevant concepts would make the paper more crowded, less focused and overall harder to read.

---

### Official Review · Reviewer_ejzP · 2022-10-24

**Confidence:** 3
**Correctness:** 4
**Technical Novelty And Significance:** 4
**Empirical Novelty And Significance:** 4
**Recommendation:** 8

**Clarity, Quality, Novelty And Reproducibility:**

The paper is well-written; the algorithm appears easy to implement; the modifications are novel to my knowledge.

**Strength And Weaknesses:**

The authors proposed a scalable approximation to the Laplace method, which is always of interest.  The proposed method is easy to implement, and demonstrates competitive performance.  The authors provided new connections to the sample-then-optimize formulation.

As for limitations, it is easy to nitpick (e.g., the limitation of Laplace method, the conjugate Gaussian model, or the linearized approximation), but I don't think these are important concerns that must be addressed in a single paper, or a single thread of literature, and the applicable scope of the proposed method is quite clear.

**Summary Of The Paper:**

This work proposes a scalable linearized Laplace procedure which can be applied to realistic BNN models.  The algorithm can be viewed as an extension of the sample-then-optimize procedure that allows for normalization layers, and hyperparameter selection is also implemented a an E-step.  The authors evaluated the method on ResNet-18 and U-Net models, demonstrating improvements on predictive performance and quality of uncertainty estimates.

**Summary Of The Review:**

Nice work with methodological and empirical contributions.

---

> ### Author Response · Authors · 2022-11-10
> **Thank you for the insightful review!**
>
> We thank the reviewer for their time reading our paper and for their kind words.

---

### Official Review · Reviewer_MCAU · 2022-10-25

**Confidence:** 3
**Correctness:** 3
**Technical Novelty And Significance:** 3
**Empirical Novelty And Significance:** 3
**Recommendation:** 8

**Clarity, Quality, Novelty And Reproducibility:**

I am not an expert in neural network uncertainty quantification, so I cannot comment on the novelty of the proposed approach in that space.  Overall, the proposed approach feels like a collection of clever tricks, each of which may not be groundbreaking, collectively allow the linearization + Laplace approximation to scale to very large datasets and neural networks.  For the most part the writing is clear, aside from some notational issues described above.  The code is available on an anonymous github repo.  I did not run the code, but it looks well-commented and well-documented.

**Strength And Weaknesses:**

Strengths:
* The approach is interesting and combines a lot of disparate ideas to make this the linearization + Laplace approximation scalable.  Some of these ideas may be of (minor) independent interest.
* The empirical runtime and uncertainty quantification results are impressive, and show a benefit to this approach for real networks.
* The writing was usually clear (but see below), and the results were presented clearly.

Weaknesses:
* In a few places the presentation was hard to follow.  For example, the rightmost plot in Figure 3 shows different layer-wise precisions, but the exposition only discusses learning a single precision (controlled by $\alpha$).  I can imagine how it is straightforward to extend to different layer-wise $\alpha$s, but that should be clarified in the main text.  In Section 4 , $\bar{w}$ is used without definition (presumably the optimized weights).  Then, $\bar{\theta}$ is used (e.g., in equation (15)) without definition.  Finally, the acronym MCDO is introduced _after_ references to Figure 6 and Table 2, where that acronym is used.  Similarly "DIP" is used in that Figure and Table, but never defined.
* The reported metrics are all good, but one of the most interpretable measures of uncertainty quantification is simply coverage, and it would be nice to report empirical coverage at different posterior credible levels on some held out data.
* I found the results in Figure 5 difficult to interpret.  Namely, it is not obvious to me how any Bayesian uncertainty quantification should perform when some datapoints are corrupted.  Would the true posterior (perhaps with a learned prior) outperform approximate posteriors in terms of log likelihood on test data?  I'm not sure that that is true, and so I'm not sure what the relative ordering of the different uncertainty quantifications methods in Figure 5 says about their "quality".

**Summary Of The Paper:**

This paper presents a number of tricks to scale linearization + Laplace approximations to large neural networks with the goal of performing uncertainty quantification.  The standard approach is to train a neural network and then post-hoc linearize the network, and use a Laplace approximation to obtain that the posterior should be Gaussian.  In this posterior, a regularization term (corresponding to the prior) appears, and this term should be fit using maximum likelihood.  To do so requires some computationally expensive numerical matrix algebra, and so the present paper instead proposes some sampling approaches (which in turn can be turned back into optimization problems) to more efficiently fit the prior and obtain samples from the posterior.  The authors apply this approach to a number of networks and datasets, finding that it can obtain better uncertainty estimates faster than existing methods (particularly in the case of simultaneously estimating uncertainty at several points).

**Summary Of The Review:**

Overall, I found the paper to be a collection of interesting tricks that scale an existing approach to larger neural networks and datasets.  The presentation was mostly clear, and the presented approach seems like it will be useful.

---

> ### Author Response · Authors · 2022-11-10
> **Thank you for the insightful feedback!**
>
> We thank the reviewer for taking the time to read our paper, and for their feedback!
>
> **Presentation**
>
> > For example, the rightmost plot in Figure 3 shows different layer-wise precisions, but the exposition only discusses learning a single precision (controlled by $\alpha$). I can imagine how it is straightforward to extend to different layer-wise αs, but that should be clarified in the main text.
>
> Thanks for the catch! We have added an explanation of how to apply our closed form precision update to diagonal but non isotropic Gaussian priors in Appendix B.4, and referenced it from where it is used in the main text.
>
> > In Section 4, $\bar{w}$ is used without definition (presumably the optimized weights)
>
> You are correct about $\bar w$ referring to the optimised weights. However, this variable is defined in the second line of section 4, right above equation 13.
>
> > Then, $\bar{\theta}$ is used (e.g., in equation (15)) without definition.
>
> In this section, $\bar \theta$ follows the same definition from section 2. We can see how this could cause confusion and have clarified the definition in equation (15) of the updated draft.
>
> > Similarly "DIP" is used in that Figure and Table, but never defined.
>
> The use of the acronym DIP, which refers to Deep Image Prior (the technique used to train the U-net in our tomographic reconstruction tasks), was unintentional and it has been removed from the updated draft.
>
> **Evaluation Metrics**
>
> > The reported metrics are all good, but one of the most interpretable measures of uncertainty quantification is simply coverage, and it would be nice to report empirical coverage at different posterior credible levels on some held out data.
>
> We agree about the relevance of width-coverage as an easily interpretable metric of uncertainty calibration. *Please find these metrics in Appendix H of our revised draft and referenced from the main text in Section 5.*
>
> However, coverage/calibration metrics are difficult to generalise to the setting of multiple output dimensions—whereas test-LL generalises naturally—and the multiple output setting is core to our work. For this reason, we chose test-LL as the main metric to report in the body of the paper.
>
> > I found the results in Figure 5 difficult to interpret. Namely, it is not obvious to me how any Bayesian uncertainty quantification should perform when some datapoints are corrupted. Would the true posterior (perhaps with a learned prior) outperform approximate posteriors in terms of log likelihood on test data? I'm not sure that that is true, and so I'm not sure what the relative ordering of the different uncertainty quantifications methods in Figure 5 says about their "quality".
>
> We agree that robustness against datapoint corruption is a misspecified problem—and that we ought not to expect the true posterior (under the misspecified assumptions) to outperform all approximate posteriors. In particular, if we were to write out a correct (well-specified) model for the corruption, we expect the posterior under that model to outperform the 'true posterior' under misspecification. However, robustness under corruption, a task motivated by real-world applications, is also perhaps the most common benchmark within the NN uncertainty quantification literature (see for example references [1 - 6]). We believe our readers (and likely other reviewers) will wish to see this experiment included.
>
> Please let us know if you have any additional suggestions on our draft!
>
> [1] https://arxiv.org/abs/1906.02530
>
> [2] https://arxiv.org/abs/2006.08437
>
> [3] https://arxiv.org/abs/2006.10108
>
> [4] https://arxiv.org/abs/2005.07186
>
> [5] https://arxiv.org/abs/2010.14689
>
> [6] https://arxiv.org/abs/2207.07411

---

### Official Review · Reviewer_d9Fu · 2022-10-26

**Confidence:** 4
**Clarity, Quality, Novelty And Reproducibility:** 1. Novelty is somehow limited. The ma…
**Correctness:** 3
**Technical Novelty And Significance:** 3
**Empirical Novelty And Significance:** 3
**Recommendation:** 6

**Strength And Weaknesses:**

Strengths:
The paper is well-written.
The approach is easy to understand and presented well in the paper.
The authors applied their method to some tasks to demonstrate the validity of the proposed method.
The theoretical analysis is strong.

Weaknesses:
1. Novelty is somehow limited. The main contribution is that the authors simply apply EM algorithm to scale inference and hyperparameter selection in Gaussian linear regression. It would be more novel if the EM algorithm was improved.
2. Experiments across multiple datasets and with larger models would make the results more convincing.
3. The ResNet-18 is also a very small model. Why not try the method on larger models? Is it because of the difficulty to implement?


**Summary Of The Paper:**

The computational cost associated with Bayesian linear models constrains this method’s application to small networks, small output spaces and small datasets. Moreover, the method is sensitive to the choice of regularisation strength for the linear model. This paper overcomes these limitations by introducing a scalable sample-based Bayesian inference method for conjugate Gaussian multi-output linear models, together with a matching method for hyperparameter selection. The experimental results demonstrate the contributions.

**Summary Of The Review:**

I am concern about the novelty of this paper.

The experiments are also suggested over larger models.

---

> ### Author Response · Authors · 2022-11-10
> **Thank you for your review, and insightful comments!**
>
> We thank the reviewer for their time reading our paper, and for their feedback!
>
> **Novelty**
>
> Our paper starts (in section 2) by describing the multi-output linear Gaussian regression model, and with an EM algorithm for inference in this setting. However, as the ultimate paragraph of this section outlines, this "simple EM" algorithm does not scale to large-data and large-model settings due to both the E and M step incurring a cubic cost.
>
> The core contribution of our paper (as outlined in section 1) is the introduction of a stochastic EM algorithm that does scale to millions of parameters and millions of observations, where both the E-step and the M-step are novel. In particular:
> 1. We use a stochastic version of the E-step, where sampling is performed using a novel objective. We show empirically that the novel objective yields an improvement of magnitude roughly equivalent to increasing batch size by a factor of 16 (Figure 2) compared to existing objectives. The E-step is thus novel.
> 2. We propose a sample-based M step. Specifically, we derive a closed-form fixed-point update for the regularisation parameter being optimised. This 'trick' is vital for the method’s performance, and is, in our understanding, novel.
>
> The reviewer doubts the novelty of the paper and states that the paper would be more novel if the EM algorithm was improved. As evidenced by our experimental results and our theory, the novel E and M steps from points 1 and 2 above do indeed result in a *much* improved EM algorithm for the multi-output Gaussian regression model.
>
> Furthermore, in the context of linearised Laplace, the specific forms of the E and M steps we choose are vital. This is because the sampling-based estimators we propose require only individual vector-Jacobian products—cost constant in the output dimension—as opposed to existing methods which require instantiating the full Jacobian matrix explicitly, at a cost of output-dimension-many vector-Jacobian products—cost linear in the output dimension.
>
> **Scale**
>
> The reviewer asks for experiments with multiple datasets and larger models. We would like to again highlight that our work:
> 1. Applies linearised Laplace to the largest dataset tackled by any previous work on linearised Laplace (CIFAR100). Previous approaches have only attempted a dataset with lower output dim (m=10 for CIFAR10), as without our sampling-based inference which depends only on vector-Jacobian products, it is computationally intractable to even assemble the Jacobian.
> 2. Applies full-covariance linearised Laplace on a model (ResNet18) much larger than any in previous literature.
> In the context of linearised Laplace literature, this pushes the boundaries on both fronts.
>
> To illustrate the complexity of linearised Laplace in the above setting: with ResNet18 on CIFAR100, we have $d \approx 11\mathrm{e}{+6}$ parameters, $n=5\mathrm{e}{+4}$ data points, and $m=100$ classes. This gives a Hessian that has either $1.21\mathrm{e}{+14}$  ($d^2$) or $2.5\mathrm{e}{+13}$ ($(nm)^2$) entries; stored as float32, these are 0.5 petabytes and 0.1 petabytes respectively. Inverting these matrices and computing their determinants present further challenges.
>
> **Overall**
>
> We hope that in light of this reply, the reviewer might reconsider their concern.

---

### Comment · Area_Chair_yEGU · 2022-11-15
**Please engage before the author-reviewer discussion closes**

Dear authors and reviewers,

The first phase of the discussion period is about to close on November 18.

For authors, please make sure to submit your rebuttal by the deadline. Leave some time for the reviewers to read it and respond while you are still allowed to further engage with them. Interactions between authors and reviewers are very important for the quality of the review process, so please make sure to engage.

For reviewers, please try to acknowledge and respond to the authors' rebuttal while the discussion period is still open for them to further interact with you.

Thank you for your participation in the review process!

Best,
The AC

---

> ### Author Response · Authors · 2022-11-15
> **Summary of updates to the draft**
>
> Thank you, AC, for starting the discussion.
>
> Following the reviewers' suggestions, here is a list of the updates we made to our draft:
>
> 1. We have added an explanation of how to apply our closed-form precision update to diagonal but non-isotropic Gaussian priors in Appendix B.4, and referenced it from where it is used in the main text.
>
> 2. We have clarified the definition of $\bar \theta$ in equation (15).
>
> 3. We have removed all uses of the acronym "DIP".
>
> 4. We have computed the "credible interval size vs coverage" predictive distribution calibration metric for our two major experiments and plotted them in Appendix H of our revised draft. We reference the plots from the main text in Section 5. We will try to have these plots feature in the main text in the camera-ready version of the paper since we think they are very illustrative.
>
>
> We are eager to hear the reviewers' thoughts on our rebuttal.
>
> Best,
>
> The authors.

---

### Decision · Program_Chairs · 2023-01-20

**Decision:**

Accept: poster

**Justification For Why Not Higher Score:**

The paper is well-written and the experiments are well done. It uses a combination of clever tricks to scale the Laplace method, which may not be groundbreaking individually but are effective when combined. For this reason, I do not recommend a higher score.

**Justification For Why Not Lower Score:**

Experiments show convincing results on previously computationally intractable inference problems.

**Metareview: Summary, Strengths And Weaknesses:**

All four reviewers unanimously recommend accepting the paper (6-8-8-8). They agree that the paper successfully connects several interesting ideas to produce a scalable approximation to the Laplace method. The experiments in the paper show results for inference settings that existing methods cannot handle, demonstrating the proposed method's usefulness. The reviewers also agree that the paper is well-written and somewhat easy to understand. The authors have made clarifications and added further experiments during the revision process.

**Note From Pc:**

if the above contains the word "oral" or "spotlight" please see: "oral" presentation means -> notable-top-5% and "spotlight" means -> notable-top-25%. As stated in our emails, we are disassociating presentation type from AC recommendations